# Risk-Sensitive Reinforcement Learning: Near-Optimal Risk-Sample Tradeoff in Regret

**Yingjie Fei**[1]   **Zhuoran Yang**[2]   **Yudong Chen**[3]   **Zhaoran Wang**[1]   **Qiaomin Xie**[3]

[1] Northwestern University; yf275@cornell.edu, zhaoranwang@gmail.com
[2] Princeton University; zy6@princeton.edu
[3] Cornell University; {yudong.chen, qiaomin.xie}@cornell.edu

## Abstract

We study risk-sensitive reinforcement learning in episodic Markov decision processes with unknown transition kernels, where the goal is to optimize the total reward under the risk measure of exponential utility. We propose two provably efficient model-free algorithms, Risk-Sensitive Value Iteration (RSVI) and Risk-Sensitive Q-learning (RSQ). These algorithms implement a form of risk-sensitive optimism in the face of uncertainty, which adapts to both risk-seeking and risk-averse modes of exploration. We prove that RSVI attains an $\tilde{O}\big(\lambda(|\beta|H^2)\cdot\sqrt{H^3S^2AT}\big)$ regret, while RSQ attains an $\tilde{O}\big(\lambda(|\beta|H^2)\cdot\sqrt{H^4SAT}\big)$ regret, where $\lambda(u) = (e^{3u}-1)/u$ for $u > 0$. In the above, $\beta$ is the risk parameter of the exponential utility function, $S$ the number of states, $A$ the number of actions, $T$ the total number of timesteps, and $H$ the episode length. On the flip side, we establish a regret lower bound showing that the exponential dependence on $|\beta|$ and $H$ is unavoidable for any algorithm with an $\tilde{O}(\sqrt{T})$ regret (even when the risk objective is on the same scale as the original reward), thus certifying the near-optimality of the proposed algorithms. Our results demonstrate that incorporating risk awareness into reinforcement learning necessitates an exponential cost in $|\beta|$ and $H$, which quantifies the fundamental tradeoff between risk sensitivity (related to aleatoric uncertainty) and sample efficiency (related to epistemic uncertainty). To the best of our knowledge, this is the first regret analysis of risk-sensitive reinforcement learning with the exponential utility.

## 1   Introduction

Risk-sensitive reinforcement learning (RL) concerns learning to act in a dynamic environment while taking into account risks that arise during the learning process. Effective management of risks in RL is critical to many real-world applications such as autonomous driving [32], real-time strategy games [56], financial investment [44], etc. In neuroscience, risk-sensitive RL has been applied to model human behaviors in decision making [46, 52].

In this paper, we consider risk-sensitive RL with the exponential utility [34] under episodic Markov decision processes (MDPs) with unknown transition kernels. Informally, the agent aims to maximize a risk-sensitive objective function of the form

$$V = \frac{1}{\beta} \log \left\{ \mathbb{E}e^{\beta R} \right\}, \tag{1}$$

where $R$ is the total reward the agent receives, and $\beta \neq 0$ is a real-valued parameter that controls risk preference of the agent; see Equation (2) for a formal definition of $V$. The objective $V$ admits the Taylor expansion $V = \mathbb{E}[R] + \frac{\beta}{2}\text{Var}(R) + O(\beta^2)$. It can be seen that for $\beta > 0$ the agent

is risk-seeking (favoring high uncertainty in $R$), for $\beta < 0$ the agent is risk-averse (favoring low uncertainty in $R$), and a larger $|\beta|$ implies higher risk-sensitivity. When $\beta \to 0$, the agent tends to be risk-neutral and the objective reduces to the expected reward objective $V = \mathbb{E}[R]$ standard in RL. Therefore, the risk-sensitive objective in (1) covers the entire spectrum of risk sensitivity by varying $\beta$. In addition, the formulation (1) is closely related to RL with constraints. For example, a negative risk parameter $\beta$ controls the tail of a risk distribution so as to mitigate the chance of receiving a total reward $R$ that is excessively low. We refer to [42, Section 2.1] for an in-depth discussion of this connection.

The challenge of risk-sensitive RL lies both in the non-linearity of the objective function and in designing a risk-aware exploration mechanism. In particular, as we elaborate in Section 2.2, the non-linear objective function (1) induces a non-linear Bellman equation. Classical RL algorithms are inappropriate in this setting, as their design crucially relies on the linearity of Bellman equations. On the other hand, effective exploration has been well known to be crucial to RL algorithm design, yet it is not clear how to design an algorithm that efficiently explores uncertain environments while at the same time adapting to the risk-sensitive objective (1) of agents with different risk parameter $\beta$.

To address these difficulties, we propose two model-free algorithms, Risk-Sensitive Value Iteration (RSVI) and Risk-Sensitive Q-learning (RSQ). Specifically, RSVI is a batch algorithm and RSQ is an online algorithm; both families of batch and online algorithms see broad applications in practice. We demonstrate in Section 3 that our proposed algorithms implement a form of risk-sensitive optimism for exploration. Importantly, the exact implementation of optimism depends on both the magnitude and the sign of the risk parameter, and therefore applies to both risk-seeking and risk-averse modes of learning. Letting $\lambda(u) = (e^{3u} - 1)/u$ for $u > 0$, we prove that RSVI attains an $\tilde{O}\big(\lambda(|\beta|H^2) \cdot \sqrt{H^3 S^2 AT}\big)$ regret, and RSQ achieves an $\tilde{O}\big(\lambda(|\beta|H^2) \cdot \sqrt{H^4 SAT}\big)$ regret. Here, $S$ and $A$ are the numbers of states and actions, respectively, $T$ is the total number of timesteps, and $H$ is the length of each episode. These regret bounds interpolate across different regimes of risk sensitivity and subsume existing results under the risk-neutral setting. Compared with risk-neutral RL (corresponding to $\beta \to 0$), our general regret bounds feature an exponential dependency on $|\beta|$ and $H$, even though the risk-sensitive objective (1) is on the same scale as the total reward; see Figure 1 for a plot of the exponential factor $\lambda(|\beta|H^2)$. Complementarily, we prove a lower bound showing that such an exponential dependency is inevitable for any algorithm and thus certifies the near-optimality of the proposed algorithms. To the best of our knowledge, our work provides the first regret analysis of risk-sensitive RL with the exponential utility.

Our upper and lower bounds demonstrate the fundamental tradeoff between risk sensitivity and sample efficiency in RL.[1] Broadly speaking, risk sensitivity is associated with *aleatoric* uncertainty, which originates from the inherent randomness of state transition, actions and rewards, whereas sample efficiency is associated with *epistemic* uncertainty, which arises from imperfect knowledge of the environment/system and can be reduced by more exploration [20, 24]. These two notions of uncertainty are usually decoupled in the regret analysis of risk-neutral RL—in particular, using the expected reward as the objective effectively suppresses the aleatoric uncertainty. In risk-sensitive RL, we establish that there is a fundamental connection and tradeoff between these two forms of uncertainty: the risk-seeking and risk-averse regimes both incur an exponential cost in $|\beta|$ and $H$ on the regret, whereas the regret is polynomial in $H$ in the risk-neutral regime.

**Our contributions.** The contributions of our work can be summarized as follows:

- We consider the problem of risk-sensitive RL with the exponential utility. We propose two provably efficient model-free algorithms, namely RSVI and RSQ, that implement risk-sensitive optimism in the face of uncertainty;

- We provide regret analysis for both algorithms over the entire spectrum of risk parameter $\beta$. As $\beta \to 0$, we show that our results recover the existing regret bounds in the risk-neutral setting;

- We provide a lower bound result that certifies the near-optimality of our upper bounds and reveals a fundamental tradeoff between risk sensitivity and sample complexity.

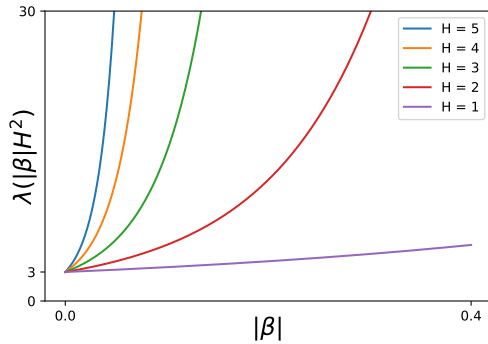

Figure 1: Scaling of $\lambda(|\beta|H^2)$ in risk sensitivity $|\beta|$ for different values of episode length $H$.

**Related work.** RL with risk-sensitive utility functions have been studied in several work. The work [45] proposes TD(0) and Q-learning-style algorithms that transform temporal differences instead of cumulative rewards, and proves their convergence. Risk-sensitive RL with a general family of utility functions is studied in [52], which also proposes a Q-learning algorithm with convergence guarantees. The work of [28] studies a risk-sensitive policy gradient algorithm, though with no theoretical guarantees. We remark that while substantial work has been devoted to designing risk-sensitive RL algorithms and proving their convergence, the issues of exploration, sample efficiency and regret bounds have rarely been studied. Our work narrows this gap in the literature by studying regret bounds of model-free algorithms for risk-sensitive RL.

The exponential utility has also been been investigated in the more classical setting of MDPs. Following the seminal work of [34], this line of work includes [7, 9–11, 14, 21, 25, 29, 30, 33, 43, 48, 51, 58, 61]. Note that these papers impose more restrictive assumptions and study different types of results than ours. Specifically, they assume known transition kernels or access to simulators, and they do not conduct finite-time or finite-sample analysis. Another related direction to ours is RL with risk/safety constraints studied by [1, 2, 16–19, 26, 27, 49, 54, 59, 62], and readers are also referred to [31] for an excellent survey on this topic. Compared to our work, that line of work focuses on constrained RL problems with different risk criteria. Other related problems include risk-sensitive games [5, 6, 8, 15, 35, 37, 40, 57], and risk-sensitive bandits [13, 22, 23, 42, 50, 53, 55, 60, 63]. Bandit problems are special cases of the RL problem that we investigate, with both the number of states and episode length being equal to one. As such, both our settings and results are more general than those obtained in bandit problems.

**Notations.** For a positive integer $n$, let $[n] := \{1, 2, \ldots, n\}$. For two non-negative sequences $\{a_i\}$ and $\{b_i\}$, we write $a_i \lesssim b_i$ if there exists a universal constant $C > 0$ such that $a_i \leq Cb_i$ for all $i$. We write $a_i \asymp b_i$ if $a_i \lesssim b_i$ and $b_i \lesssim a_i$. We use $\tilde{O}(\cdot)$ to denote $O(\cdot)$ while hiding logarithmic factors.

## 2 Problem setup

### 2.1 Episodic MDPs and risk-sensitive objective

We consider the setting of episodic MDPs, denoted by $\text{MDP}(\mathcal{S}, \mathcal{A}, H, \mathcal{P}, \mathcal{R})$, where $\mathcal{S}$ is the set of possible states, $\mathcal{A}$ is the set of possible actions, $H$ is the length of each episode, and $\mathcal{P} = \{P_h\}_{h \in [H]}$ and $\mathcal{R} = \{r_h\}_{h \in [H]}$ are the sets of state transition kernels and reward functions, respectively. In particular, for each $h \in [H]$, $P_h(\cdot \,|\, s, a)$ is the distribution of the next state if action $a$ is taken in state $s$ at step $h$. We assume that $\mathcal{S}$ and $\mathcal{A}$ are finite discrete spaces, and let $S = |\mathcal{S}|$ and $A = |\mathcal{A}|$ denote their cardinalities. We assume that the agent does not have access to $\{P_h\}$ and that each $r_h : \mathcal{S} \times \mathcal{A} \to [0, 1]$ is a deterministic function.

An agent interacts with an episodic MDP as follows. At the beginning of each episode, an initial state $s_1$ is chosen arbitrarily by the environment. In each step $h \in [H]$, the agent observes a state $s_h \in \mathcal{S}$, chooses an action $a_h \in \mathcal{A}$, and receives a reward $r_h(s_h, a_h)$. The MDP then transitions into a new

state $s_{h+1} \sim P_h(\cdot \mid s_h, a_h)$. We use the convention that the episode terminates when a state $s_{H+1}$ at step $H + 1$ is reached, at which the agent does not take an action and receives no reward.

A policy $\pi = \{\pi_h\}_{h \in [H]}$ of an agent is a sequence of functions $\pi_h : \mathcal{S} \to \mathcal{A}$, where $\pi_h(s)$ is the action that the agent takes in state $s$ at step $h$ of an episode. For each $h \in [H]$, we define the value function $V_h^\pi : \mathcal{S} \to \mathbb{R}$ of a policy $\pi$ as the expected value of cumulative rewards the agent receives under a risk measure of exponential utility by executing policy $\pi$ starting from an arbitrary state at step $h$. Specifically, we have

$$V_h^\pi(s) := \frac{1}{\beta} \log \left\{ \mathbb{E} \left[ \exp \left( \beta \sum_{h'=h}^{H} r_{h'}(s_{h'}, \pi_{h'}(s_{h'})) \right) \,\middle|\, s_h = s \right] \right\}, \tag{2}$$

for each $(h, s) \in [H] \times \mathcal{S}$. Here $\beta \neq 0$ is the risk parameter of the exponential utility: $\beta > 0$ corresponds to a risk-seeking value function, $\beta < 0$ corresponds to a risk-averse value function, and as $\beta \to 0$ the agent tends to be risk-neutral and we recover the classical value function $V_h^\pi(s) = \mathbb{E}[\sum_{h=1}^{H} r_h(s_h, \pi_h(s_h)) \mid s_h = s]$ in RL. The goal of the agent is to find a policy $\pi$ such that $V_1^\pi(s)$ is maximized for all state $s \in \mathcal{S}$. Note the logarithm and rescaling by $1/\beta$ in the above definition, which puts the objective $V_1^\pi(s)$ on the same scale as the total reward; this scaling property is made formal in Lemma 1 below.

## 2.2 Bellman equations and regret

We further define the action-value function $Q_h^\pi : \mathcal{S} \times \mathcal{A} \to \mathbb{R}$, which gives the expected value of the risk measured by the exponential utility when the agent starts from an arbitrary state-action pair at step $h$ and follows policy $\pi$ afterwards; that is,

$$Q_h^\pi(s, a) := \frac{1}{\beta} \log \left\{ \exp(\beta \cdot r_h(s, a)) \mathbb{E} \left[ \exp \left( \beta \sum_{h'=h+1}^{H} r_{h'}(s_{h'}, a_{h'}) \right) \,\middle|\, s_h = s, a_h = a \right] \right\},$$

for all $(h, s, a) \in [H] \times \mathcal{S} \times \mathcal{A}$. The Bellman equation associated with policy $\pi$ is given by

$$\begin{aligned} Q_h^\pi(s, a) &= r_h(s, a) + \frac{1}{\beta} \log \left\{ \mathbb{E}_{s' \sim P_h(\cdot \mid s, a)} \left[ \exp \left( \beta \cdot V_{h+1}^\pi(s') \right) \right] \right\}, \\ V_h^\pi(s) &= Q_h^\pi(s, \pi_h(s)), \qquad V_{H+1}^\pi(s) = 0, \end{aligned} \tag{3}$$

which holds for all $(s, a) \in \mathcal{S} \times \mathcal{A}$.

Under some mild regularity conditions, there always exists an optimal policy $\pi^*$ which gives the optimal value $V_h^*(s) = \sup_\pi V_h^\pi(s)$ for all $(h, s) \in [H] \times \mathcal{S}$ [7]. The Bellman optimality equation is given by

$$\begin{aligned} Q_h^*(s, a) &= r_h(s, a) + \frac{1}{\beta} \log \left\{ \mathbb{E}_{s' \sim P_h(\cdot \mid s, a)} \left[ \exp \left( \beta \cdot V_{h+1}^*(s') \right) \right] \right\}, \\ V_h^*(s) &= \max_{a \in \mathcal{A}} Q_h^*(s, a), \qquad V_{H+1}^*(s) = 0. \end{aligned} \tag{4}$$

This equation implies that the optimal policy $\pi^*$ is the greedy policy with respect to the optimal action-value function $\{Q_h^*\}_{h \in [H]}$. Hence, to find the optimal policy $\pi^*$, it suffices to estimate the optimal action-value function. We note that both Bellman equations (3) and (4) are non-linear in the value and action-value functions due to non-linearity of the exponential utility. This is in contrast with their linear risk-neutral counterparts.

Under the episodic MDP setting, the agent aims to learn the optimal policy by interacting with the environment throughout a set of episodes. For each $k \geq 1$, let us denote by $s_1^k$ the initial state chosen by the environment and $\pi^k$ the policy chosen simultaneously by the agent at the beginning of episode $k$. The difference in values between $V_1^{\pi^k}(s_1^k)$ and $V_1^*(s_1^k)$ measures the expected regret or the sub-optimality of the agent in episode $k$. After $K$ episodes, the total regret for the agent is

$$\text{Regret}(K) := \sum_{k \in [K]} \left[ V_1^*(s_1^k) - V_1^{\pi^k}(s_1^k) \right]. \tag{5}$$

We record the following simple worst-case upper bounds on the value functions and regret.

**Lemma 1.** *For any* $(h, s, a) \in \mathcal{S} \times \mathcal{A} \times [H]$, *policy* $\pi$ *and risk parameter* $\beta \neq 0$, *we have*

$$0 \leq V_h^\pi(s) \leq H \quad and \quad 0 \leq Q_h^\pi(s, a) \leq H. \tag{6}$$

*Consequently, for each* $K \geq 1$, *all policy sequences* $\pi^1, \dots, \pi^K$ *and any* $\beta \neq 0$, *we have*

$$0 \leq \text{Regret}(K) \leq KH. \tag{7}$$

*Proof.* Recall the assumption that the reward functions $\{r_h\}$ are bounded in $[0, 1]$. The lower bounds are immediate by definition. For the upper bound, we have $V_h^\pi(s) \leq \frac{1}{\beta} \log \{\mathbb{E} [\exp (\beta H)]\} = H$. Upper bounds for $Q_h^\pi$ and the regret follow similarly. $\square$

While straightforward, the above lemma highlights an important point: the risk and regret are on the same scale as the reward. In particular, the upper bounds above are *independent* of $\beta$ and *linear* in the horizon length $H$—the same as in the standard MDP setting—because the $\log$ and $\exp$ functions in the definition of the objective function (2) cancel with each other in the worst case. Therefore, the exponential dependence of the regret on $|\beta|$ and $H$, which we establish below in Section 4, is not merely a consequence of scaling but rather is inherent in the risk-sensitive setting.

## 3 Algorithms

The non-linearity of the Bellman equations, discussed in Section 2.2, creates challenges in algorithmic design. In particular, standard model-free algorithms such as least-squares value iteration (LSVI) and Q-learning are no longer appropriate since they specialize to the risk-neutral setting with linear Bellman equations. In this section, we present risk-sensitive LSVI and Q-learning algorithms that adapt to both the non-linear Bellman equations and any valid risk parameter $\beta$.

### 3.1 Risk-Sensitive Value Iteration

We first present Risk-Sensitive Value Iteration (RSVI) in Algorithm 1. Algorithm 1 is inspired by LSVI-UCB of [39], which is in turn motivated by the idea of LSVI [12, 47] and the classical value-iteration algorithm. Like LSVI-UCB, Algorithm 1 applies the Upper Confidence Bound (UCB) by incorporating a bonus term to value estimates of state-action pairs, which therefore implements the principle of Optimism in the Face of Uncertainty (OFU) [36].

**Mechanism of Algorithm 1.** The algorithm mainly consists of the value estimation step (Line 6–13) and the policy execution step (Line 14–18). In Line 7, the algorithm computes the intermediate value $w_h$ by a least-squares update

$$w_h \leftarrow \underset{w \in \mathbb{R}^{SA}}{\text{argmin}} \sum_{\tau \in [k-1]} \left[ e^{\beta[r_h(s_h^\tau, a_h^\tau) + V_{h+1}(s_{h+1}^\tau)]} - w^\top \phi(s_h^\tau, a_h^\tau) \right]^2. \tag{8}$$

Here, $\{(s_h^\tau, a_h^\tau, s_{h+1}^\tau)\}_{\tau \in [k-1]}$ are accessed from the dataset $\mathcal{D}_h$ for each $h \in [H]$, and $\phi(\cdot, \cdot)$ denotes the canonical basis in $\mathbb{R}^{SA}$. Line 7 can be efficiently implemented by computing sample means of $e^{\beta[r_h(s,a) + V_{h+1}(s')]}$ over those state-action pairs that the algorithm has visited. Therefore, it can also be interpreted as estimating the sample means of exponentiated $Q$-values under visitation measures induced by the transition kernels $\{P_h\}$. This is a typical feature of the family of batch algorithms, to which Algorithm 1 belongs. Then, in Line 10, the algorithm uses the intermediate value $w_h$ to compute the estimate $Q_h$, by adding/subtracting bonus $b_h$ and thresholding the sum/difference at $e^{\beta(H-h+1)}$, depending on the sign of $\beta$. It is not hard to see that the logarithmic-exponential transformation in Line 10 conforms and adapts to the non-linearity in Bellman equations (3) and (4). In addition, the thresholding operator ensures that the estimated action-value function $Q_h$ of step $h$ stays in the range $[0, H-h+1]$ and so does the estimated value function $V_h$ in Line 11. This is to enforce the estimates $Q_h$ and $V_h$ to be on the same scale as the optimal $Q_h^*$ and $V_h^*$.

Besides the logarithmic-exponential transformation, another distinctive feature of Algorithm 1 is the way the bonus term $b_h > 0$ is incorporated in Line 10. At first sight, it might appear counter-intuitive to *subtract* $b_h$ from $w_h$ when $\beta < 0$. We demonstrate next that subtracting bonus when $\beta < 0$ in fact implements the idea of OFU in a risk-sensitive fashion.

---

**Algorithm 1** RSVI

---

**Input:** number of episodes $K \in \mathbb{Z}_{>0}$, confidence level $\delta \in (0, 1]$, and risk parameter $\beta \neq 0$

1:   $Q_h(s, a) \leftarrow H - h + 1$ and $N_h(s, a) \leftarrow 0$ for all $(h, s, a) \in [H] \times \mathcal{S} \times \mathcal{A}$
2:   $Q_{H+1}(s, a) \leftarrow 0$ for all $(s, a) \in \mathcal{S} \times \mathcal{A}$
3:   Initialize datasets $\{\mathcal{D}_h\}$ as empty
4:   **for** episode $k = 1, \ldots, K$ **do**
5:      $V_{H+1}(s) \leftarrow 0$ for each $s \in \mathcal{S}$
6:      **for** step $h = H, \ldots, 1$ **do**                              ▷ *value estimation*
7:         Update $w_h$ via Equation (8)
8:         **for** $(s, a) \in \mathcal{S} \times \mathcal{A}$ such that $N_h(s, a) \geq 1$ **do**
9:             $b_h(s, a) \leftarrow c_\gamma \left| e^{\beta H} - 1 \right| \sqrt{\frac{S \log(2SAT/\delta)}{N_h(s, a)}}$ for some universal constant $c_\gamma > 0$
10:           $Q_h(s, a) \leftarrow \begin{cases} \frac{1}{\beta} \log \left[ \min\{e^{\beta(H-h+1)}, w_h(s, a) + b_h(s, a)\} \right], & \text{if } \beta > 0; \\ \frac{1}{\beta} \log \left[ \max\{e^{\beta(H-h+1)}, w_h(s, a) - b_h(s, a)\} \right], & \text{if } \beta < 0 \end{cases}$
11:           $V_h(s) \leftarrow \max_{a' \in \mathcal{A}} Q_h(s, a')$
12:         **end for**
13:      **end for**
14:      **for** step $h = 1, \ldots, H$ **do**                                  ▷ *policy execution*
15:         Take action $a_h \leftarrow \operatorname{argmax}_{a \in \mathcal{A}} Q_h(s_h, a)$ and observe $r_h(s_h, a_h)$ and $s_{h+1}$
16:         $N_h(s_h, a_h) \leftarrow N_h(s_h, a_h) + 1$
17:         Insert $(s_h, a_h, s_{h+1})$ into $\mathcal{D}_h$
18:      **end for**
19: **end for**

---

**Risk-Sensitive Upper Confidence Bound.** For the purpose of illustration, let us consider a "promising" state $s^+ \in \mathcal{S}$ at step $h$ that allows us to transition to states $\{s'\}$ in the next step with high values $\{V_{h+1}(s')\}$ regardless of actions taken. This means that the intermediate value $w_h(s^+, \cdot) \propto \sum_{s'} e^{\beta \cdot V_{h+1}(s')}$ tends to be *small,* given that $\beta < 0$ and $\{V_{h+1}(s')\}$ are large. By subtracting a positive $b_h$ from $w_h$, we obtain an even smaller quantity $w_h(s^+, \cdot) - b_h(s^+, \cdot)$. We can then deduce that $Q_h(s^+, \cdot) \approx \frac{1}{\beta} \log[w_h(s^+, \cdot) - b_h(s^+, \cdot)]$ is *larger* compared to $\frac{1}{\beta} \log[w_h(s^+, \cdot)]$ which does not incorporate bonus, since the logarithmic function is monotonic and again $\beta < 0$ (we ignore thresholding for the moment). Therefore, subtracting bonus serves as a UCB for $\beta < 0$. Since the exact form of the UCB depends on both the magnitude and sign of $\beta$ (as shown in Lines 9 and 10), we name it Risk-Sensitive Upper Confidence Bound (RS-UCB) and this results in what we call Risk-Sensitive Optimism in the Face of Uncertainty (RS-OFU).

### 3.2   Risk-Sensitive Q-learning

Although Algorithm 1 is model-free, it requires storage of historical data $\{\mathcal{D}_h\}$ and computation over them (Line 7). A more efficient class of algorithms is Q-learning algorithms, which update Q values in an online fashion as each state-action pair is encountered. We therefore propose Risk-Sensitive Q-learning (RSQ) and formally describe it in Algorithm 2.

**Mechanism of Algorithm 2.** Algorithm 2 is based on Q-learning with UCB studied in the work of [38] and we use the same learning rates therein

$$\alpha_t := \frac{H + 1}{H + t} \tag{9}$$

for every integer $t \geq 1$. Similar to Algorithm 1, Algorithm 2 consists of the policy execution step (Line 6) and value estimation step (Lines 9–11). Line 9 updates the intermediate value $w_h$ in an online fashion, in constrast with the batch update in Line 7 of Algorithm 1, and Algorithm 2 can thus be seen as an online algorithm. Line 10 then applies the same logarithmic-exponential transform to the intermediate value and bonus as in Algorithm 1. Note the similar way we use the bonus term $b_t$ in estimating $Q$-values in Line 10 of Algorithm 2 as in Line 10 of Algorithm 1. Algorithm 2 therefore also implements RS-UCB and follows the principle of RS-OFU.

**Algorithm 2** RSQ

---

**Input:** number of episodes $K \in \mathbb{Z}_{>0}$, confidence level $\delta \in (0, 1]$, learning rates $\{\alpha_t\}$ and risk parameter $\beta \neq 0$

1: $Q_h(s, a), V_h(s, a) \leftarrow H - h + 1$ and $N_h(s, a) \leftarrow 0$ for all $(h, s, a) \in [H] \times \mathcal{S} \times \mathcal{A}$
2: $Q_{H+1}(s, a), V_{H+1}(s, a) \leftarrow 0$ for all $(s, a) \in \mathcal{S} \times \mathcal{A}$
3: **for** episode $k = 1, \ldots, K$ **do**
4:      Receive the initial state $s_1$
5:      **for** step $h = 1, \ldots, H$ **do**
6:          Take action $a_h \leftarrow \text{argmax}_{a' \in \mathcal{A}} Q_h(s_h, a')$, and observe $r_h(s_h, a_h)$ and $s_{h+1}$
7:          $t = N_h(s_h, a_h) \leftarrow N_h(s_h, a_h) + 1$
8:          $b_t \leftarrow c \left| e^{\beta H} - 1 \right| \sqrt{\frac{H \log(SAT/\delta)}{t}}$ for some sufficiently large universal constant $c > 0$
9:          $w_h(s_h, a_h) \leftarrow (1 - \alpha_t) e^{\beta \cdot Q_h(s_h, a_h)} + \alpha_t e^{\beta[r_h(s_h, a_h) + V_{h+1}(s_{h+1})]}$
10:        $Q_h(s_h, a_h) \leftarrow \begin{cases} \frac{1}{\beta} \log \left[ \min\{e^{\beta(H-h+1)}, w_h(s_h, a_h) + \alpha_t b_t\} \right], & \text{if } \beta > 0; \\ \frac{1}{\beta} \log \left[ \max\{e^{\beta(H-h+1)}, w_h(s_h, a_h) - \alpha_t b_t\} \right], & \text{if } \beta < 0 \end{cases}$
11:          $V_h(s_h) \leftarrow \max_{a' \in \mathcal{A}} Q_h(s_h, a')$
12:      **end for**
13: **end for**

---

**Comparisons of Algorithms 1 and 2.** It is interesting to compare the bonuses used in Algorithms 1 and 2. The bonuses in both algorithms depend on the risk parameter $\beta$ through a common factor $\left| e^{\beta H} - 1 \right|$. A careful analysis (see our proofs in appendices) on the bonuses and the value estimation steps reveals that the effective bonuses added to the estimated value function is proportional to $\frac{e^{|\beta|H} - 1}{|\beta|}$. This means that the more risk-seeking/averse an agent is (or the larger $|\beta|$ is), the larger bonus it needs to compensate for its uncertainty over the environment. Such risk sensitivity of the bonus is also reflected in the regret bounds; see Theorems 1 and 2 below. Also, it is not hard to see that both algorithms have polynomial time and space complexities in $S$, $A$, $K$ and $H$. Moreover, thanks to its online update procedure, Algorithm 2 is more efficient than Algorithms 1 in both time and space complexities, since it does not require storing historical data (in particular, $\{\mathcal{D}_h\}$ of Algorithm 1) nor computing statistics based on them for value estimation.

## 4 Main results

In this section, we first present regret bounds for Algorithms 1 and 2, and then we complement the results with a lower bound on regret that any algorithm has to incur.

### 4.1 Regret upper bounds

The following theorem gives an upper bound for regret incurred by Algorithm 1. Let $T := KH$ be the total number of timesteps for which an algorithm is run, and recall the function $\lambda(u) := (e^{3u} - 1)/u$.

**Theorem 1.** *For any $\delta \in (0, 1]$, with probability at least $1 - \delta$, the regret of Algorithm 1 is bounded by*

$$\text{Regret}(K) \lesssim \lambda(|\beta|H^2) \cdot \sqrt{H^3 S^2 AT \log^2(2SAT/\delta)}.$$

The proof is given in Appendix C. We see that the result of Theorem 1 adapts to both risk-seeking ($\beta > 0$) and risk-averse ($\beta < 0$) settings through a common factor of $\lambda(|\beta|H^2)$.

As $\beta \to 0$, the setting of risk-sensitive RL tends to that of standard and risk-neutral RL, and we have an immediate corollary to Theorem 1 as a precise characterization.

**Corollary 1.** *Under the setting of Theorem 1 and when $\beta \to 0$, with probability at least $1 - \delta$, the regret of Algorithm 1 is bounded by*

$$\text{Regret}(K) \lesssim \sqrt{H^3 S^2 AT \log^2(2SAT/\delta)}.$$

*Proof.* The result follows from Theorem 1 and the fact that $\lim_{\beta \to 0} \lambda(|\beta|H^2) = 3$.     $\square$

The result in Corollary 1 recovers the regret bound of [4, Theorem 2] under the standard RL setting and is nearly optimal compared to the minimax rates presented in [3, Theorems 1 and 2]. Corollary 1 also reveals that Theorem 1 interpolates between the risk-sensitive and risk-neutral settings.

Next, we give a regret upper bound for Algorithm 2 in the following theorem.

**Theorem 2.** *For any $\delta \in (0, 1]$, with probability at least $1 - \delta$ and when $T$ is sufficiently large, the regret of Algorithm 2 is bounded by*

$$\text{Regret}(K) \lesssim \lambda(|\beta|H^2) \cdot \sqrt{H^4 SAT \log(SAT/\delta)}.$$

The proof is given in Appendix E. Similarly to Theorem 1, Theorem 2 also covers both risk-seeking and risk-averse settings via the same factor $\lambda(|\beta|H^2)$, which gives the risk-neutral bound when $\beta \to 0$ as shown in the following.

**Corollary 2.** *Under the setting of Theorem 2 and when $\beta \to 0$, with probability at least $1 - \delta$, the regret of Algorithm 2 is bounded by*

$$\text{Regret}(K) \lesssim \sqrt{H^4 SAT \log(SAT/\delta)}.$$

The proof follows the same reasoning as in that of Corollary 1. According to Corollary 2, the regret upper bound for Algorithm 2 matches the nearly optimal result in [38, Theorem 2] under the risk-neutral setting. As such, Theorems 1 and 2 strictly generalizes the existing nearly optimal regret bounds (up to polynomial factors).

The crux of the proofs of both Theorems 1 and 2 lies in a local linearization argument for the non-linear Bellman equations and non-linear updates of the algorithms, in which action-value and value functions are related by a logarithmic-exponential transformation. Although logarithmic and exponential functions are not Lipschitz globally, we show that they are locally Lipschitz in the domain of our interest, and their combined local Lipschitz factors turn out to be the exponential factors in the theorems. Once the Bellman equations and algorithm estimates are linearized, we can apply standard techniques in RL to obtain the final regret. It is noteworthy that, as suggested by [38], the regret bounds in Theorems 1 and 2 can automatically be translated into sample complexity bounds in the probably approximately correct (PAC) setting, which did not previously exist even given access to a simulator.

In the risk-sensitive setting where $\beta$ is bounded away from 0, our regret bounds of Theorems 1 and 2 depend exponentially in the horizon length $H$ and the risk sensitivity $|\beta|$. In what follows, we argue that such exponential dependence is unavoidable.

## 4.2 Regret lower bound

We now present a fundamental lower bound on the regret, which complements the upper bounds in Theorems 1 and 2.

**Theorem 3.** *If $|\beta|(H - 1)$ and $K$ are sufficiently large, the regret of any policy obeys*

$$\text{Regret}(K) \gtrsim \lambda(|\beta|(H - 1)/6) \cdot \sqrt{HT}.$$

The proof is given in Appendix F. In the proof, we construct an MDP that can be reduced to a bandit problem. We then show that any bandit algorithm has to incur an expected regret, in terms of the logarithmic-exponential objective, that grows as predicted in Theorem 3.

Theorem 3 shows that the exponential dependence on the $|\beta|$ and $H$ in Theorems 1 and 2 is essentially indispensable. In addition, it features a sub-linear dependence on $T$ through the $\tilde{O}(\sqrt{T})$ factor. In view of Theorem 3, therefore, both Theorems 1 and 2 are nearly optimal in their dependence on $\beta$, $H$ and $T$. One should contrast Theorem 3 with Lemma 1, which shows that the worst-case regret is linear in $H$ and $T$. Such a linear regret can be attained by any trivial algorithm that does not learn at all. In sharp contrast, in order to achieve the optimal $\sqrt{T}$ scaling (which by standard arguments implies a finite sample-complexity bound), an algorithm must incur a regret that is exponential in $H$. Therefore, our results show a (perhaps surprising) tradeoff between risk sensitivity and sample efficiency.

## Broader Impact

This work contributes to the risk-awareness of machine learning and improves the way RL algorithms handle risks arising from uncertain environments. We have proposed two efficient and model-free algorithms for risk-sensitive RL with the exponential utility. We show that both algorithms follow the principle of Risk-Sensitive Optimism in the Face of Uncertainty (RS-OFU), and they achieve nearly optimal regret bounds with respect to the risk parameter, horizon length and total number of timesteps.

## Acknowledgments and Disclosure of Funding

Y. Chen is partially supported by NSF grants 1657420 and 1704828. Q. Xie is partially supported by NSF grant 1955997. The other co-authors have no funding to disclose.

## Footnotes

[1]By standard arguments, regret can be translated into sample complexity bounds and vice versa; see [38].

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
