[Supplementary Material]

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

# Appendices

## A Preliminaries

We set some notations and shorthands before the proofs. For both Algorithms 1 and 2, we let $s_h^k$, $a_h^k$, $w_h^k$, $Q_h^k$ and $V_h^k$ denote the values of $s_h$, $a_h$, $w_h$, $Q_h$ and $V_h$ in episode $k$, and we denote by $N_h^k$ the value of $N_h$ at the end of episode $k-1$. For Algorithm 1, we let $\mathcal{D}_h^k$ be the value of $\mathcal{D}_h$ at the end of episode $k-1$. Next, we introduce a simple yet powerful result.

**Fact 1.** *Consider $x, y, b \in \mathbb{R}$ such that $x \geq y$.*

(a) *if $y \geq g$ for some $g > 0$, then $\log(x) - \log(y) \leq \frac{1}{g}(x-y)$;*

(b) *Assume further that $y \geq 0$. If $b \geq 0$ and $x \leq u$ for some $u > 0$, then $e^{bx} - e^{by} \leq be^{bu}(x-y)$; if $b < 0$, then $e^{by} - e^{bx} \leq (-b)(x-y)$.*

*Proof.* The results follow from Lipschitz continuity of the functions $x \mapsto \log(x)$ and $x \mapsto e^{bx}$. $\qquad\square$

We record a simple fact about exponential factors.

**Fact 2.** *Define $\lambda_0 := \frac{e^{|\beta|H}-1}{|\beta|}$ and $\lambda_2 := e^{|\beta|(H^2+H)}$. Then we have $\lambda_0 \lambda_2 H \leq \frac{e^{3|\beta|H^2}-1}{|\beta|}$.*

## B Proof warmup for Theorem 1

First, we set some notations and definitions. Define $d := SA$, $\iota := \log(2dT/\delta)$ for a given $\delta \in (0, 1]$, and $I$ to be the $d \times d$ identity matrix. To streamline some parts of the proof, we define $\phi(s, a)$ to be a vector in $\mathbb{R}^d$ whose $(s, a)$-th entry is equal to one and other entries equal to zero (so $\phi(s, a)$ is a canonical basis of $\mathbb{R}^{SA}$). Also let $\Lambda_h^k$ be a diagonal matrix in $\mathbb{R}^{d \times d}$ with each $(s, a)$-th diagonal entry equal to $\max\{N_h^{k-1}(s, a), 1\}$. It can be seen that $\Lambda_h^k$ is positive definite. We adopt the shorthands $\phi_h^\tau := \phi(s_h^\tau, a_h^\tau)$ and $r_h^\tau := r_h(s_h^\tau, a_h^\tau)$ for $(\tau, h) \in [K] \times [H]$.

From now on, we fix a tuple $(k, h) \in [K] \times [H]$ and then fix $(s, a) \in \mathcal{S} \times \mathcal{A}$ such that $N_h^{k-1}(s, a) \geq 1$. We also fix a policy $\pi$. We set

$$w_h^\pi = e^{\beta \cdot Q_h^\pi(\cdot, \cdot)}. \tag{10}$$

It can be verified that by the definition of $\phi(s, a)$, we have

$$\begin{aligned} Q_h^\pi(s, a) &= \frac{1}{\beta} \log\left(e^{\beta \cdot Q_h^\pi(s,a)}\right) \\ &= \frac{1}{\beta} \log\left(\left\langle \phi(s, a), e^{\beta \cdot Q_h^\pi(\cdot, \cdot)}\right\rangle\right) \\ &= \frac{1}{\beta} \log\left(\langle \phi(s, a), w_h^\pi\rangle\right), \end{aligned} \tag{11}$$

as well as

$$w_h^\pi(s, a) = e^{\beta \cdot Q_h^\pi(s,a)} = \left\langle \phi(s, a), (\Lambda_h^k)^{-1} \sum_{\tau \in [k-1]} \phi_h^\tau \left[e^{\beta \cdot Q_h^\pi(s_h^\tau, a_h^\tau)}\right]\right\rangle, \tag{12}$$

where the last step follows from the definition of $\Lambda_h^k$.

Let us define

$$\begin{aligned} q_1^+ &:= \begin{cases} \langle \phi(s, a), w_h^k\rangle + b_h^k(s, a), & \text{if } \beta > 0, \\ \langle \phi(s, a), w_h^k\rangle - b_h^k(s, a), & \text{if } \beta < 0, \end{cases} \\ q_1 &:= \begin{cases} \min\{e^{\beta(H-h+1)}, q_1^+\}, & \text{if } \beta > 0, \\ \max\{e^{\beta(H-h+1)}, q_1^+\}, & \text{if } \beta < 0. \end{cases} \end{aligned}$$

By the definition of $\Lambda_h^k$ and $\phi_h^k$, observe that

$$w_h^k(s,a) = \langle \phi(s,a), w_h^k \rangle = \left\langle \phi(s,a), (\Lambda_h^k)^{-1} \sum_{\tau \in [k-1]} \phi_h^\tau \left[ e^{\beta[r_h^\tau + V_{h+1}^k(s_{h+1}^\tau)]} \right] \right\rangle. \tag{13}$$

Define

$$G_0 := (Q_h^k - Q_h^\pi)(s,a) = \frac{1}{\beta} \log\{q_1\} - \frac{1}{\beta} \log\{\langle \phi(s,a), w_h^\pi \rangle\}, \tag{14}$$

and our goal is to derive lower and upper bounds for $G_0$. From Equation (14), we have

$$G_0 = \frac{1}{\beta} \log\{q_1\} - \frac{1}{\beta} \log\left\{ \left\langle \phi(s,a), (\Lambda_h^k)^{-1} \sum_{\tau \in [k-1]} \phi_h^\tau \left[ e^{\beta \cdot Q_h^\pi(s_h^\tau, a_h^\tau)} \right] \right\rangle \right\}$$

$$= \frac{1}{\beta} \log\{q_1\} - \frac{1}{\beta} \log\left\{ \left\langle \phi(s,a), (\Lambda_h^k)^{-1} \sum_{\tau \in [k-1]} \phi_h^\tau \left[ \mathbb{E}_{s' \sim P_h(\cdot \mid s_h^\tau, a_h^\tau)} e^{\beta[r_h^\tau + V_{h+1}^\pi(s')]} \right] \right\rangle \right\}$$

$$=: \frac{1}{\beta} \log\{q_1\} - \frac{1}{\beta} \log\{q_3\}.$$

The first step above holds by Equation (12), and the second step follows from Equation (3). In order to control $G_0$, we define an intermediate quantity

$$q_2 := \left\langle \phi(s,a), (\Lambda_h^k)^{-1} \sum_{\tau \in [k-1]} \phi_h^\tau \left[ \mathbb{E}_{s' \sim P_h(\cdot \mid s_h^\tau, a_h^\tau)} e^{\beta[r_h^\tau + V_{h+1}^k(s')]} \right] \right\rangle;$$

in words, $q_2$ replaces the quantity $V_{h+1}^\pi$ in $q_3$ by $V_{h+1}^k$. It can be seen that

$$G_0 = G_1 + G_2, \tag{15}$$

where

$$G_1 := \frac{1}{\beta} \log\{q_1\} - \frac{1}{\beta} \log\{q_2\},$$

$$G_2 := \frac{1}{\beta} \log\{q_2\} - \frac{1}{\beta} \log\{q_3\}. \tag{16}$$

Note that $G_0$, $G_1$ and $G_2$ are all well-defined, according to the following result.

**Lemma 2.** *We have* $q_i \in [\min\{1, e^{\beta(H-h+1)}\}, \max\{1, e^{\beta(H-h+1)}\}]$ *for* $i \in [3]$.

*Proof.* We prove the result by focusing on $q_1$. By the definitions of $\Lambda_h^k$ and $\phi$, the $(s,a)$-th entry of the vector $(\Lambda_h^k)^{-1} \sum_{\tau \in [k-1]} \phi_h^\tau \cdot u_h^\tau$ equals $\frac{1}{N_h^{k-1}(s,a)} \sum_{\tau \in [k-1]} u_h^\tau \cdot \mathbb{I}\{(s_h^\tau, a_h^\tau) = (s,a)\}$ for any sequence $\{u_h^\tau\}_{\tau \in [k-1]}$. Then, the result follows from the fact that $e^{\beta[r_h^\tau + V_{h+1}^k(s')]} \in [\min\{1, e^{\beta(H-h)}\}, \max\{1, e^{\beta(H-h)}\}]$ for $(\tau, s') \in [K] \times \mathcal{S}$ and the definition of $q_1$. $\square$

Therefore, we have the following equivalent form of Equation (14):

$$(Q_h^k - Q_h^\pi)(s,a) = G_1 + G_2. \tag{17}$$

Thanks to the identity (17), our goal is now to control $G_1$ and $G_2$, which is done in the following lemma.

**Lemma 3.** *For all* $(k, h, s, a) \in [K] \times [H] \times \mathcal{S} \times \mathcal{A}$ *that satisifies* $N_h^{k-1}(s,a) \geq 1$, *there exist universal constants* $c_1, c_\gamma > 0$ *(where* $c_\gamma$ *is used in Line 9 of Algorithm 1) such that*

$$0 \leq G_1 \leq c_1 \cdot \frac{e^{|\beta|H} - 1}{|\beta|} \cdot d\sqrt{\iota} \sqrt{\phi(s,a)^\top (\Lambda_h^k)^{-1} \phi(s,a)}$$

*with probability at least* $1 - \delta/2$. *Furthermore, if* $V_{h+1}^k(s') \geq V_{h+1}^\pi(s')$ *for all* $s' \in \mathcal{S}$, *then we have*

$$0 \leq G_2 \leq e^{|\beta|H} \cdot \mathbb{E}_{s' \sim P_h(\cdot \mid s,a)}[V_{h+1}^k(s') - V_{h+1}^\pi(s')].$$

*Proof.* **Case** $\beta > 0$**.** To control $G_1$, we note that $N_h^{k-1}(s,a) = \phi(s,a)^\top (\Lambda_h^k)^{-1}\phi(s,a)$ and by Equation (13) we can compute

$$\left| q_1^+ - q_2 - b_h^k(s,a) \right|$$

$$= \left| \left\langle \phi(s,a), (\Lambda_h^k)^{-1} \sum_{\tau \in [k-1]} \phi_h^\tau \left[ e^{\beta[r_h^\tau + V_{h+1}^k(s_{h+1}^\tau)]} - \mathbb{E}_{s' \sim P_h(\cdot \mid s_h^\tau, a_h^\tau)} e^{\beta[r_h^\tau + V_{h+1}^k(s')]} \right] \right\rangle \right|$$

$$= \left| \frac{1}{N_h^{k-1}(s,a)} \sum_{(s,a,s^+) \in \mathcal{D}_h^{k-1}} e^{\beta[r_h(s,a) + V_{h+1}^k(s^+)]} - \mathbb{E}_{s' \sim P_h(\cdot \mid s,a)} e^{\beta[r_h(s,a) + V_{h+1}^k(s')]} \right|$$

$$\leq \frac{1}{N_h^{k-1}(s,a)} \sum_{(s,a,s^+) \in \mathcal{D}_h^{k-1}} \left| e^{\beta[r_h(s,a) + V_{h+1}^k(s^+)]} - \mathbb{E}_{s' \sim P_h(\cdot \mid s,a)} e^{\beta[r_h(s,a) + V_{h+1}^k(s')]} \right|$$

$$\leq \frac{1}{N_h^{k-1}(s,a)} \sum_{t \in [N_h^{k-1}(s,a)]} c' \left| e^{\beta H} - 1 \right| \sqrt{\frac{S\iota}{t}}$$

$$\leq \frac{1}{N_h^{k-1}(s,a)} \int_{t \in [0, N_h^{k-1}(s,a)]} c' \left| e^{\beta H} - 1 \right| \sqrt{\frac{S\iota}{t}} \, dt$$

$$= \frac{1}{N_h^{k-1}(s,a)} \cdot c \left| e^{\beta H} - 1 \right| \sqrt{S\iota \cdot N_h^{k-1}(s,a)}$$

$$= c \left| e^{\beta H} - 1 \right| \sqrt{S\iota} \cdot \sqrt{\phi(s,a)^\top (\Lambda_h^k)^{-1} \phi(s,a)},$$

where the fourth step holds by Lemma 6, and the last step holds by the definition of $\Lambda_h^k$; in the above, $c' > 0$ is a universal constant and $c = 2c'$. If we choose $c_\gamma = c$ in the definition of $b_h^k(s,a)$ in Line 9 of Algorithm 1, we have

$$0 \leq q_1^+ - q_2 \leq 2c \cdot \left| e^{\beta H} - 1 \right| \sqrt{S\iota} \cdot \sqrt{\phi(s,a)^\top (\Lambda_h^k)^{-1} \phi(s,a)}.$$

Therefore, we have $q_1 \geq q_2$, and thus $G_1 \geq 0$, by the first inequality above, the definition of $q_1$ and Lemma 2 (in particular, $q_2 \leq e^{\beta(H-h+1)}$). By Lemma 2 and Fact 1(a) (with $g = 1$, $x = q_1$ and $y = q_2$), we have

$$G_1 \leq \frac{1}{\beta}(q_1 - q_2) \leq \frac{1}{\beta}(q_1^+ - q_2),$$

which together with the second inequality displayed above implies the desired upper bound on $G_1$.

Now we control the term $G_2$. For $\beta > 0$, it is not hard to see that the assumption $V_{h+1}^k(s') \geq V_{h+1}^\pi(s')$ for all $s' \in \mathcal{S}$ implies that $q_2 \geq q_3$ and therefore $G_2 \geq 0$. We also have

$$G_2 \leq \frac{1}{\beta}(q_2 - q_3)$$

$$\leq e^{\beta H} \left\langle \phi(s,a), (\Lambda_h^k)^{-1} \sum_{\tau \in [k-1]} \phi_h^\tau \left[ \mathbb{E}_{s' \sim P_h(\cdot \mid s_h^\tau, a_h^\tau)} [V_{h+1}^k(s') - V_{h+1}^\pi(s')] \right] \right\rangle$$

$$= e^{|\beta| H} \mathbb{E}_{s' \sim P_h(\cdot \mid s,a)} [V_{h+1}^k(s') - V_{h+1}^\pi(s')],$$

where the first step holds by Fact 1(a) (with $g = 1$, $x = q_2$, and $y = q_3$) and the fact that $q_2 \geq q_3 \geq 1$ (with the last inequality suggested by Lemma 2), and the second step holds by Fact 1(b) (with $b = \beta$, $x = r_h^\tau + V_{h+1}^k(s)$, and $y = r_h^\tau + V_{h+1}^\pi(s)$) and $H \geq r_h^\tau + V_{h+1}^k(s) \geq r_h^\tau + V_{h+1}^\pi(s) \geq 0$.

**Case** $\beta < 0$**.** Similar to the case of $\beta > 0$, we have

$$\left| q_1^+ - q_2 + b_h^k(s,a) \right|$$

$$\leq c \cdot \left| e^{\beta H} - 1 \right| \sqrt{S\iota} \cdot \sqrt{\phi(s,a)^\top (\Lambda_h^k)^{-1} \phi(s,a)}.$$

If we choose $c_\gamma = c$ in the definition of $b_h^k(s,a)$ in Line 9 of Algorithm 1, the above equation implies

$$0 \leq q_2 - q_1^+ \leq 2c \cdot \left| e^{\beta H} - 1 \right| \sqrt{S\iota} \cdot \sqrt{\phi(s,a)^\top (\Lambda_h^k)^{-1} \phi(s,a)}.$$

Therefore, we have $q_1 \leq q_2$, and thus $G_1 \geq 0$, by the first inequality displayed above, the definition of $q_1$ and Lemma 2 (in particular, $q_2 \geq e^{\beta(H-h+1)}$). By Lemma 2 and Fact 1(a) (with $g = e^{\beta H}$, $x = q_2$ and $y = q_1$), we further have

$$
\begin{aligned}
G_1 &= \frac{1}{(-\beta)} \left( \log\{q_2\} - \log\{q_1\} \right) \\
&\leq \frac{e^{-\beta H}}{|\beta|} (q_2 - q_1) \\
&\leq \frac{e^{-\beta H}}{|\beta|} (q_2 - q_1^+),
\end{aligned}
$$

which together with the second inequality displayed above and the fact that $\left| e^{\beta H} - 1 \right| = 1 - e^{\beta H}$ implies the desired upper bound on $G_1$.

Next we control $G_2$. The assumption $V_{h+1}^k(s') \geq V_{h+1}^\pi(s')$ for all $s' \in \mathcal{S}$ implies that $q_2 \leq q_3$ and therefore $G_2 \geq 0$. We also have

$$
\begin{aligned}
G_2 &= \frac{1}{(-\beta)} \left( \log\{q_3\} - \log\{q_2\} \right) \\
&\leq \frac{e^{-\beta H}}{(-\beta)} (q_3 - q_2) \\
&\leq e^{|\beta|H} \left\langle \phi(s,a), (\Lambda_h^k)^{-1} \sum_{\tau \in [k-1]} \phi_h^\tau \left[ \mathbb{E}_{s' \sim P_h(\cdot \mid s_h^\tau, a_h^\tau)}[V_{h+1}^k(s') - V_{h+1}^\pi(s')] \right] \right\rangle \\
&= e^{|\beta|H} \mathbb{E}_{s' \sim P_h(\cdot \mid s,a)}[V_{h+1}^k(s') - V_{h+1}^\pi(s')],
\end{aligned}
$$

where the second step holds by Fact 1(a) (with $g = e^{\beta H}$, $x = q_3$, and $y = q_2$) and the fact that $q_3 \geq q_2 \geq e^{\beta H}$ (with the last inequality suggested by Lemma 2), and the third step holds by Fact 1(b) (with $b = \beta$, $x = r_h^\tau + V_{h+1}^k(s)$, and $y = r_h^\tau + V_{h+1}^\pi(s)$) and $r_h^\tau + V_{h+1}^k(s) \geq r_h^\tau + V_{h+1}^\pi(s) \geq 0$.

The proof is hence completed. $\qquad \square$

The next lemma establishes the dominance of $Q_h^k$ over $Q_h^*$.

**Lemma 4.** *On the event of Lemma 3, we have $Q_h^k(s,a) \geq Q_h^\pi(s,a)$ for all $(k, h, s, a) \in [K] \times [H] \times \mathcal{S} \times \mathcal{A}$.*

*Proof.* For the purpose of the proof, we set $Q_{H+1}^\pi(s,a) = Q_{H+1}^*(s,a) = 0$ for all $(s,a) \in \mathcal{S} \times \mathcal{A}$. We fix a tuple $(k, s, a) \in [K] \times \mathcal{S} \times \mathcal{A}$ and use strong induction on $h$. The base case for $h = H+1$ is satisfied since $(Q_{H+1}^k - Q_{H+1}^\pi)(s,a) = 0$ for $k \in [K]$ by definition. Now we fix an $h \in [H]$ and assume that $0 \leq (Q_{h+1}^k - Q_{h+1}^*)(s,a)$. Moreover, by the induction assumption we have

$$
V_{h+1}^k(s) = \max_{a' \in \mathcal{A}} Q_{h+1}^k(s, a') \geq \max_{a' \in \mathcal{A}} Q_{h+1}^\pi(s, a') \geq V_{h+1}^\pi(s). \tag{18}
$$

We also assume that $(s,a)$ satisfies $N_h^{k-1}(s,a) \geq 1$, since otherwise $Q_h^k(s,a) = H - h + 1 \geq Q_h^\pi(s,a)$ and we are done. This assumption and Equation (18) together imply $G_2 \geq 0$ by Lemma 3. We also have $G_1 \geq 0$ on the event of Lemma 3. Therefore, it follows that $(Q_h^k - Q_h^\pi)(s,a) \geq 0$ by Equation (17). The induction is completed and so is the proof. $\qquad \square$

Lemma 4 leads to an immediate and important corollary.

**Lemma 5.** *For any $\delta \in (0,1]$, with probability at least $1 - \delta/2$, we have $V_h^k(s) \geq V_h^\pi(s)$ for all $(k, h, s) \in [K] \times [H] \times \mathcal{S}$.*

*Proof.* The result follows from Lemma 4 and Equation (18). $\qquad \square$

## B.1 Supporting lemmas

We first present a concentration result.

**Lemma 6.** *Define*

$$\bar{\mathcal{V}}_{h+1} := \left\{ \bar{V}_{h+1} : \mathcal{S} \to \mathbb{R} \mid \forall s \in \mathcal{S}, \ \bar{V}_{h+1}(s) \in [\min\{e^{\beta(H-h)}, 1\}, \max\{e^{\beta(H-h)}, 1\}] \right\}.$$

*There exists a universal constant $c > 0$ such that with probability $1 - \delta$, we have*

$$\left| e^{\beta[r_h(s_h^k, a_h^k) + \bar{V}(s_{h+1}^k)]} - \mathbb{E}_{s' \sim P_h(\cdot | s_h^k, a_h^k)} e^{\beta[r_h(s_h^k, a_h^k) + \bar{V}(s')]} \right| \leq c \left| e^{\beta H} - 1 \right| \sqrt{\frac{S\iota}{N_h^k(s, a)}}$$

*for all $(k, h, s, a) \in [K] \times [H] \times \mathcal{S} \times \mathcal{A}$ and all $\bar{V} \in \bar{\mathcal{V}}_{h+1}$.*

*Proof.* The proof follows the same reasoning as [4, Lemma 12]. $\qquad\square$

The next few lemmas help control $\sum_{k \in [K]} (\phi_h^k)^\top (\Lambda_h^k)^{-1} \phi_h^k$.

**Lemma 7** ([39, Lemma D.2]). *Let $\{\phi_t\}_{t \geq 0}$ be a bounded sequence in $\mathbb{R}^d$ satisfying $\sup_{t \geq 0} \|\phi_t\| \leq 1$. Let $\Lambda_0 \in \mathbb{R}^{d \times d}$ be a positive definite matrix with $\lambda_{\min}(\Lambda_0) \geq 1$. For any $t \geq 0$, we define $\Lambda_t := \Lambda_0 + \sum_{i \in [t]} \phi_i \phi_i^\top$. Then, we have*

$$\log \left[ \frac{\det(\Lambda_t)}{\det(\Lambda_0)} \right] \leq \sum_{i \in [t]} \phi_i^\top \Lambda_{i-1}^{-1} \phi_i \leq 2 \log \left[ \frac{\det(\Lambda_t)}{\det(\Lambda_0)} \right].$$

**Lemma 8.** *Recall the definitions of $\phi_h^k$ and $\Lambda_h^k$. For any $h \in [H]$, we have*

$$\sum_{k \in [K]} (\phi_h^k)^\top (\Lambda_h^k)^{-1} \phi_h^k \leq 2d\iota,$$

*where $\iota = \log(2dT/\delta)$*

*Proof.* Define $\Gamma_h^k := \lambda I + \sum_{\tau \in [k-1]} \phi_h^\tau (\phi_h^\tau)^\top$ with $\lambda = 1$. It is not hard to see that by the definition of $\Lambda_h^k$ we have $\Lambda_h^k \preceq \Gamma_h^k$ for $h \in [H]$. Since $\lambda_{\min}(\Gamma_h^k) \geq 1$ and $\|\phi_h^k\| \leq 1$ for all $(k, h) \in [K] \times [H]$, by Lemma 7 we have for any $h \in [H]$ that

$$\sum_{k \in [K]} (\phi_h^k)^\top (\Lambda_h^k)^{-1} \phi_h^k \leq \sum_{k \in [K]} (\phi_h^k)^\top (\Gamma_h^k)^{-1} \phi_h^k \leq 2 \log \left[ \frac{\det(\Gamma_h^{k+1})}{\det(\Gamma_h^1)} \right].$$

Furthermore, note that $\|\Gamma_h^{k+1}\| = \|\lambda I + \sum_{\tau \in [k]} \phi_h^k (\phi_h^k)^\top\| \leq \lambda + k$. This implies

$$\sum_{k \in [K]} (\phi_h^k)^\top (\Lambda_h^k)^{-1} \phi_h^k \leq 2d \log \left[ \frac{\lambda + k}{\lambda} \right] \leq 2d\iota,$$

as desired. $\qquad\square$

## C  Proof of Theorem 1

Define $\delta_h^k := V_h^k(s_h^k) - V_h^{\pi_k}(s_h^k)$, and $\zeta_{h+1}^k := \mathbb{E}_{s' \sim P_h(\cdot | s_h^k, a_h^k)}[V_{h+1}^k(s') - V_{h+1}^{\pi_k}(s')] - \delta_{h+1}^k$. For any $(k, h) \in [K] \times [H]$, we have

$$\delta_h^k = (Q_h^k - Q_h^{\pi_k})(s_h^k, a_h^k)$$

$$\leq c_1 \cdot \frac{e^{|\beta| H} - 1}{|\beta|} \cdot \sqrt{S\iota} \sqrt{\phi(s_h^k, a_h^k)^\top (\Lambda_h^k)^{-1} \phi(s_h^k, a_h^k)}$$

$$+ e^{|\beta| H} \cdot \mathbb{E}_{s' \sim P_h(\cdot | s_h^k, a_h^k)}[V_{h+1}^k(s') - V_{h+1}^{\pi_k}(s')]$$

$$= c_1 \cdot \frac{e^{|\beta|H} - 1}{|\beta|} \cdot \sqrt{S\iota} \sqrt{\phi(s_h^k, a_h^k)^\top (\Lambda_h^k)^{-1} \phi(s_h^k, a_h^k)}$$

$$+ e^{|\beta|H} (\delta_{h+1}^k + \zeta_{h+1}^k). \tag{19}$$

In the above equation, the first step holds by the construction of Algorithm 1 and the definition of $V_h^{\pi_k}$ in Equation (3); the second step is a consequence of combining Equation (17) as well as Lemmas 3 and 5; the last step follows from the definitions of $\delta_h^k$ and $\zeta_{h+1}^k$.

Noting that $V_{H+1}^k(s) = V_{H+1}^{\pi_k}(s) = 0$ and the fact that $\delta_{h+1}^k + \zeta_{h+1}^k \geq 0$ implied by Lemma 5, we can continue by expanding the recursion in Equation (19) and get

$$\delta_1^k \leq \sum_{h \in [H]} e^{(|\beta|H)h} \zeta_{h+1}^k$$

$$+ c_1 \cdot \frac{e^{|\beta|H} - 1}{|\beta|} \cdot \sum_{h \in [H]} e^{(|\beta|H)(h-1)} \sqrt{S\iota} \sqrt{\phi(s_h^k, a_h^k)^\top (\Lambda_h^k)^{-1} \phi(s_h^k, a_h^k)}. \tag{20}$$

Therefore, we have

$$\text{Regret}(K) = \sum_{k \in [K]} \left[ (V_1^* - V_1^{\pi_k})(s_1^k) \right]$$

$$\leq \sum_{k \in [K]} \delta_1^k$$

$$\leq e^{|\beta|H^2} \sum_{k \in [K]} \sum_{h \in [H]} \zeta_{h+1}^k$$

$$+ c_1 \cdot \frac{e^{|\beta|H} - 1}{|\beta|} \cdot e^{|\beta|H^2} \cdot \sqrt{S\iota} \sum_{k \in [K]} \sum_{h \in [H]} \sqrt{\phi(s_h^k, a_h^k)^\top (\Lambda_h^k)^{-1} \phi(s_h^k, a_h^k)}, \tag{21}$$

where the second step holds by Lemma 5 with $\pi$ therein set to the optimal policy, and in the last step we applied Equation (20) along with the Cauchy-Schwarz inequality.

We proceed to control the two terms in Equation (21). Since the construction of $V_h^k$ is independent of the new observation $s_h^k$ in episode $k$, we have that $\{\zeta_{h+1}^k\}$ is a martingale difference sequence satisfying $|\zeta_h^k| \leq 2H$ for all $(k, h) \in [K] \times [H]$. By the Azuma-Hoeffding inequality, we have for any $t > 0$,

$$\mathbb{P} \left( \sum_{k \in [K]} \sum_{h \in [H]} \zeta_{h+1}^k \geq t \right) \leq \exp \left( -\frac{t^2}{2T \cdot H^2} \right).$$

Hence, with probability $1 - \delta/2$, there holds

$$\sum_{k \in [K]} \sum_{h \in [H]} \zeta_{h+1}^k \leq \sqrt{2TH^2 \cdot \log(2/\delta)} \leq 2H\sqrt{T\iota}, \tag{22}$$

where $\iota = \log(2dT/\delta)$. For the second term in Equation (21), we apply Lemma 8 and the Cauchy-Schwarz inequality to obtain

$$\sum_{k \in [K]} \sum_{h \in [H]} \sqrt{\phi(s_h^k, a_h^k)^\top (\Lambda_h^k)^{-1} \phi(s_h^k, a_h^k)}$$

$$\leq \sum_{h \in [H]} \sqrt{K} \sqrt{\sum_{k \in [K]} \phi(s_h^k, a_h^k)^\top (\Lambda_h^k)^{-1} \phi(s_h^k, a_h^k)}$$

$$\leq H\sqrt{2dK\iota}. \tag{23}$$

Plugging Equations (22) and (23) back to Equation (21) yields

$$\text{Regret}(K) \leq e^{|\beta|H^2} \cdot 2H\sqrt{T\iota} + c_1 \cdot \frac{e^{|\beta|H} - 1}{|\beta|} \cdot e^{|\beta|H^2} \cdot H\sqrt{2dSK\iota^2}$$

$$\leq (c_1 + 2) \cdot \frac{e^{|\beta|H} - 1}{|\beta|} \cdot e^{|\beta|H^2} \cdot \sqrt{2dHST\iota^2},$$

where the last step holds since $\frac{e^{|\beta|H} - 1}{|\beta|} \geq H$. The proof is completed in view of Fact 2 and the identity $d = SA$.

## D   Proof warmup for Theorem 2

Recall the learning rates $\{\alpha_t\}$ defined in Equation (9). Define the quantities

$$\alpha_t^0 := \prod_{j=1}^{t} (1 - \alpha_j), \qquad \alpha_t^i := \alpha_i \prod_{j=i+1}^{t} (1 - \alpha_j) \qquad (24)$$

for integers $i, t \geq 1$. By convention, we set $\alpha_t^0 = 1$ and $\sum_{i \in [t]} \alpha_t^i = 0$ if $t = 0$, and $\alpha_t^i = \alpha_i$ if $t < i + 1$. Define the shorthand $\iota := \log(SAT/\delta)$ for $\delta \in (0, 1]$.

The following fact describes some key properties of the learning rates $\{\alpha_t\}$.

**Fact 3.** *The following properties hold for $\alpha_t^i$.*

(a) $\frac{1}{\sqrt{t}} \leq \sum_{i \in [t]} \frac{\alpha_t^i}{\sqrt{i}} \leq \frac{2}{\sqrt{t}}$ *for every integer $t \geq 1$.*

(b) $\max_{i \in [t]} \alpha_t^i \leq \frac{2H}{t}$ *and* $\sum_{i \in [t]} (\alpha_t^i)^2 \leq \frac{2H}{t}$ *for every integer $t \geq 1$.*

(c) $\sum_{t=i}^{\infty} \alpha_t^i = 1 + \frac{1}{H}$ *for every integer $i \geq 1$.*

(d) $\sum_{i \in [t]} \alpha_t^i = 1$ *and* $\alpha_t^0 = 0$ *for every integer $t \geq 1$, and* $\sum_{i \in [t]} \alpha_t^i = 0$ *and* $\alpha_t^0 = 1$ *for $t = 0$.*

*Proof.* The first three facts can be found in [38, Lemma 4.1], and the last one follows from direct calculation in view of Equation (24). $\qquad\square$

We also present a lemma that controls the deviation of the exponentiated value function from its expectation.

**Lemma 9.** *There exists a universal constant $c > 0$ such that for any $(k, h, s, a) \in [K] \times [H] \times \mathcal{S} \times \mathcal{A}$ and $k_1, \ldots, k_t < k$ with $t = N_h^k(s, a)$, we have*

$$\left| \frac{1}{\beta} \sum_{i \in [t]} \alpha_t^i \left[ e^{\beta[r_h(s,a) + V_{h+1}^*(s_{h+1}^{k_i})]} - \mathbb{E}_{s' \sim P_h(\cdot \mid s,a)} e^{\beta[r_h(s,a) + V_{h+1}^*(s')]} \right] \right|$$

$$\leq \frac{c \left| e^{\beta H} - 1 \right|}{|\beta|} \sqrt{\frac{H\iota}{t}}.$$

*with probability at least $1 - \delta$, and*

$$\frac{1}{|\beta|} \sum_{i \in [t]} \alpha_t^i b_i \in \left[ \frac{c \left| e^{\beta H} - 1 \right|}{|\beta|} \sqrt{\frac{H\iota}{t}}, \frac{2c \left| e^{\beta H} - 1 \right|}{|\beta|} \sqrt{\frac{H\iota}{t}} \right].$$

*Proof.* For any $(k, h, s, a) \in [K] \times [H] \times \mathcal{S} \times \mathcal{A}$, define

$$\psi(i, k, h, s, a) := e^{\beta[r_h(s,a) + V_{h+1}^*(s_{h+1}^{k_i})]} - \mathbb{E}_{s' \sim P_h(\cdot \mid s,a)} e^{\beta[r_h(s,a) + V_{h+1}^*(s')]}$$

Let us fix a tuple $(k, h, s, a) \in [K] \times [H] \times \mathcal{S} \times \mathcal{A}$. It can be seen that $\{\mathbb{I}(k_i \leq K) \cdot \psi(i, k, h, s, a)\}_{i \in [\tau]}$ for $\tau \in [K]$ is a martingale difference sequence. By the Azuma-Hoeffding inequality and a union bound over $\tau \in [K]$, we have with probability at least $1 - \delta/(HSA)$, for all $\tau \in [K]$,

$$\left| \sum_{i \in [\tau]} \alpha_\tau^i \cdot \mathbb{I}(k_i \leq K) \cdot \psi(i, k, h, s, a) \right|$$

$$\leq \frac{c \left| e^{\beta H} - 1 \right|}{2} \sqrt{\iota \sum_{i \in [\tau]} (\alpha_\tau^i)^2} \leq c \left| e^{\beta H} - 1 \right| \sqrt{\frac{H \iota}{\tau}}$$

where $c > 0$ is some universal constant, the first step holds since $r_h(s,a) + V_{h+1}^*(s') \in [0, H]$ for $s' \in \mathcal{S}$, and the last step follows from Fact 3(b). Since the above equation holds for all $\tau \in [K]$, it also holds for $\tau = t = N_h^k(s,a) \leq K$. Note that $\mathbb{I}(k_i \leq K) = 1$ for all $i \in [N_h^k(s,a)]$. Therefore, applying another union bound over $(h, s, a) \in [H] \times \mathcal{S} \times \mathcal{A}$, we have that the following holds for all $(k, h, s, a) \in [K] \times [H] \times \mathcal{S} \times \mathcal{A}$ and with probability at least $1 - \delta$:

$$\left| \sum_{i \in [t]} \alpha_\tau^i \cdot \psi(i, k, h, s, a) \right| \leq c \left| e^{\beta H} - 1 \right| \sqrt{\frac{H \iota}{t}}, \tag{25}$$

where $t = N_h^k(s,a)$. Using the fact that $r_h + V_{h+1}^* \in [0, H]$, we have

$$\left| \frac{1}{\beta} \sum_{i \in [t]} \alpha_t^i \left[ \mathbb{E}_{s' \sim \hat{P}_h^{k_i}(\cdot \mid s, a)} e^{\beta [r_h(s,a) + V_{h+1}^*(s')]} - \mathbb{E}_{s' \sim P_h(\cdot \mid s, a)} e^{\beta [r_h(s,a) + V_{h+1}^*(s')]} \right] \right|$$

$$= \left| \frac{1}{\beta} \sum_{i \in [t]} \alpha_t^i \cdot \psi(i, k, h, s, a) \right| \leq \frac{c \left| e^{\beta H} - 1 \right|}{|\beta|} \sqrt{\frac{H \iota}{t}}.$$

To prove the result for $\frac{1}{|\beta|} \sum_{i \in [t]} \alpha_t^i b_i$, we recall the definition of $\{b_t\}$ in Line 8 of Algorithm 2 and compute

$$\frac{1}{|\beta|} \sum_{i \in [t]} \alpha_t^i b_i = \frac{c \left| e^{\beta H} - 1 \right|}{|\beta|} \sum_{i \in [t]} \alpha_t^i \sqrt{\frac{H \iota}{i}}$$

$$\in \left[ \frac{c \left| e^{\beta H} - 1 \right|}{|\beta|} \sqrt{\frac{H \iota}{t}}, \frac{2c \left| e^{\beta H} - 1 \right|}{|\beta|} \sqrt{\frac{H \iota}{t}} \right]$$

where the last step holds by Fact 3(a). $\qquad \square$

We fix a tuple $(k, h, s, a) \in [K] \times [H] \times \mathcal{S} \times \mathcal{A}$ with $k_i \leq k$ being the episode in which $(s,a)$ is taken the $i$-th time at step $h$. Let us define

$$q_1^+ := \begin{cases} \alpha_t^0 e^{\beta(H-h+1)} + \sum_{i \in [t]} \alpha_t^i \left[ e^{\beta [r_h(s,a) + V_{h+1}^{k_i}(s_{h+1}^{k_i})]} + b_i \right], & \text{if } \beta > 0, \\ \alpha_t^0 e^{\beta(H-h+1)} + \sum_{i \in [t]} \alpha_t^i \left[ e^{\beta [r_h(s,a) + V_{h+1}^{k_i}(s_{h+1}^{k_i})]} - b_i \right], & \text{if } \beta < 0, \end{cases}$$

$$q_1 := \begin{cases} \min\{e^{\beta(H-h+1)}, q_1^+\}, & \text{if } \beta > 0, \\ \max\{e^{\beta(H-h+1)}, q_1^+\}, & \text{if } \beta < 0, \end{cases}$$

and

$$q_2^+ := \begin{cases} \alpha_t^0 e^{\beta(H-h+1)} + \sum_{i \in [t]} \alpha_t^i \left[ e^{\beta [r_h(s,a) + V_{h+1}^*(s_{h+1}^{k_i})]} + b_i \right], & \text{if } \beta > 0, \\ \alpha_t^0 e^{\beta(H-h+1)} + \sum_{i \in [t]} \alpha_t^i \left[ e^{\beta [r_h(s,a) + V_{h+1}^*(s_{h+1}^{k_i})]} - b_i \right], & \text{if } \beta < 0, \end{cases}$$

$$q_2 := \begin{cases} \min\{e^{\beta(H-h+1)}, q_2^+\}, & \text{if } \beta > 0, \\ \max\{e^{\beta(H-h+1)}, q_2^+\}, & \text{if } \beta < 0, \end{cases}$$

$$q_2' := \alpha_t^0 e^{\beta(H-h+1)} + \sum_{i \in [t]} \alpha_t^i \left[ e^{\beta [r_h(s,a) + V_{h+1}^*(s_{h+1}^{k_i})]} \right],$$

and

$$q_3 := \alpha_t^0 e^{\beta \cdot Q_h^*(s,a)} + \sum_{i \in [t]} \alpha_t^i \left[ \mathbb{E}_{s' \sim P_h(\cdot \mid s, a)} e^{\beta [r_h(s,a) + V_{h+1}^*(s')]} \right].$$

We have a simple fact on $q_2$ and $q_2'$.

**Fact 4.** *If $\beta > 0$, we have $q_2' \leq q_2$; if $\beta < 0$, we have $q_2' \geq q_2$.*

*Proof.* We focus on the case of $\beta > 0$. Note that $r_h(s, a) + V_{h+1}^*(s_{h+1}^{k_i}) \in [0, H - h + 1]$, which implies $e^{\beta[r_h(s,a)+V_{h+1}^*(s_{h+1}^{k_i})]} \leq e^{\beta(H-h+1)}$. We also have $\alpha_t^0, \sum_{i \in [t]} \alpha_t^i \in \{0, 1\}$ with $\alpha_t^0 + \sum_{i \in [t]} \alpha_t^i = 1$ by Fact 3(d). These together imply that $q_2' \leq e^{\beta H}$ and $q_2' - q_2^+ = -\sum_{i \in [t]} \alpha_t^i b_i \leq 0$ by definition of $b_i$ in Line 8 of Algorithm 2. Therefore, $q_2' \leq \min\{e^{\beta(H-h+1)}, q_2^+\} = q_2$. The case of $\beta < 0$ can be proved in a similar way and thus omitted. $\square$

Next, we establish a representation of the performance difference $(Q_h^k - Q_h^*)(s, a)$ using the quantities $q_1$ and $q_3$.

**Lemma 10.** *For any $(k, h, s, a) \in [K] \times [H] \times \mathcal{S} \times \mathcal{A}$, let $t = N_h^k(s, a)$ and suppose $(s, a)$ was previously taken at step $h$ of episodes $k_1, \ldots, k_t < k$. We have*

$$(Q_h^k - Q_h^*)(s, a) = \frac{1}{\beta} \log\{q_1\} - \frac{1}{\beta} \log\{q_3\}.$$

*Proof.* The Bellman optimality equation (4) implies

$$e^{\beta \cdot Q_h^*(s,a)} = e^{\beta \cdot r_h(s,a)} \left[ \mathbb{E}_{s' \sim P_h(\cdot \mid s, a)} e^{\beta \cdot V_{h+1}^*(s')} \right].$$

By Fact 3(d), we have

$$e^{\beta \cdot Q_h^*(s,a)} = \alpha_t^0 e^{\beta \cdot Q_h^*(s,a)} + \sum_{i \in [t]} \alpha_t^i e^{\beta \cdot r_h(s,a)} \left[ \mathbb{E}_{s' \sim P_h(\cdot \mid s, a)} e^{\beta \cdot V_{h+1}^*(s')} \right] = q_3$$

for each integer $t \geq 0$, and therefore

$$Q_h^*(s, a) = \frac{1}{\beta} \log\{q_3\}. \tag{26}$$

We finish the proof by combining Equation (26) and the fact that $Q_h^k(s, a) = \frac{1}{\beta} \log\{q_1\}$, which follows from Line 10 of Algorithm 2. $\square$

We define the quantities

$$\begin{aligned} G_1 &:= \frac{1}{\beta} \log\{q_1\} - \frac{1}{\beta} \log\{q_2\}, \\ G_2 &:= \frac{1}{\beta} \log\{q_2\} - \frac{1}{\beta} \log\{q_3\}, \end{aligned} \tag{27}$$

It is not hard to see that $(Q_h^k - Q_h^*)(s, a) = G_1 + G_2$ by Lemma 10. The next lemma establishes upper and lower bounds for $(Q_h^k - Q_h^*)(s, a)$.

**Lemma 11.** *For all $(k, h, s, a) \in [K] \times [H] \times \mathcal{S} \times \mathcal{A}$ such that $t = N_h^k(s, a) \geq 1$, let*

$$\gamma_t := 2 \sum_{i \in [t]} \alpha_t^i b_i \cdot \begin{cases} \frac{1}{|\beta|}, & \text{if } \beta > 0, \\ \frac{e^{-\beta H}}{|\beta|}, & \text{if } \beta < 0, \end{cases}$$

*and with probability at least $1 - \delta$ we have*

$$0 \leq (Q_h^k - Q_h^*)(s, a) \leq \alpha_t^0 H e^{|\beta|H} + \sum_{i \in [t]} \alpha_t^i e^{|\beta|H} \left[ V_{h+1}^{k_i}(s_{h+1}^{k_i}) - V_{h+1}^*(s_{h+1}^{k_i}) \right] + 2\gamma_t,$$

*where $k_1, \ldots, k_t < k$ are the episodes in which $(s, a)$ was taken at step $h$, and $\gamma_t \leq \frac{4c(e^{|\beta|H}-1)}{|\beta|} \sqrt{\frac{H\iota}{t}}$.*

*Proof.* We prove the lower bound for $(Q_h^k - Q_h^*)(s, a)$ and then use it to prove the upper bound.

**Lower bound for $Q^k - Q^*$.**

For the purpose of the proof, we set $Q_{H+1}^k(s,a) = Q_{H+1}^*(s,a) = 0$ for all $(k,s,a) \in [K] \times \mathcal{S} \times \mathcal{A}$. We fix a $(s,a) \in \mathcal{S} \times \mathcal{A}$ and use strong induction on $k$ and $h$. Without loss of generality, we assume that there exists a $(k,h)$ such that $(s,a) = (s_h^k, a_h^k)$ (that is, $(s,a)$ has been taken at some point in Algorithm 2), since otherwise $Q_h^k(s,a) = H - h + 1 \geq Q_h^*(s,a)$ for all $(k,h) \in [K] \times [H]$ and we are done. The base case for $k = 1$ and $h = H + 1$ is satisfied since $(Q_{H+1}^{k'} - Q_{H+1}^*)(s,a) = 0$ for $k' \in [K]$ by definition. We fix a $(k,h) \in [K] \times [H]$ and assume that $0 \leq (Q_{h+1}^{k_i} - Q_{h+1}^*)(s,a)$ for each $k_1, \ldots, k_t < k$ (here $t = N_h^k(s,a)$). Then we have for $i \in [t]$ that

$$V_{h+1}^{k_i}(s) = \max_{a' \in \mathcal{A}} Q_{h+1}^{k_i}(s,a') \geq \max_{a' \in \mathcal{A}} Q_{h+1}^*(s,a') = V_{h+1}^*(s).$$

Recall the quantities $G_1$ and $G_2$ defined in Equation (27). The above equation implies $G_1 \geq 0$. We also have $G_2 \geq 0$ by the fact $Q_h^*(s,a) \leq H$ and on the event of Lemma 9. Therefore, it follows that $(Q_h^k - Q_h^*)(s,a) = G_1 + G_2 \geq 0$. The induction is completed and we have proved that $0 \leq (Q_h^k - Q_h^*)(s,a)$ for all $(k,h,s,a) \in [K] \times [H] \times \mathcal{S} \times \mathcal{A}$.

**Upper bound for $Q^k - Q^*$.**

Let us fix a $(k,h,s,a) \in [K] \times [H] \times \mathcal{S} \times \mathcal{A}$. Since $0 \leq (Q_h^k - Q_h^*)(s,a)$, we have for $i \in [t]$ that

$$V_{h+1}^{k_i}(s) = \max_{a' \in \mathcal{A}} Q_{h+1}^{k_i}(s,a') \geq \max_{a' \in \mathcal{A}} Q_{h+1}^*(s,a') = V_{h+1}^*(s).$$

**Case $\beta > 0$.** We have

$$
\begin{aligned}
G_1 &= \frac{1}{\beta} \log\{q_1\} - \frac{1}{\beta} \log\{q_2\} \\
&\leq \frac{1}{\beta}(q_1 - q_2) \\
&\leq \frac{1}{\beta}(q_1^+ - q_2') \\
&\leq \frac{1}{\beta} \sum_{i \in [t]} \alpha_t^i \left[ e^{\beta[r_h(s,a) + V_{h+1}^{k_i}(s_{h+1}^{k_i})]} - e^{\beta[r_h(s,a) + V_{h+1}^*(s_{h+1}^{k_i})]} \right] + \frac{1}{\beta} \sum_{i \in [t]} \alpha_t^i b_i \\
&\leq e^{|\beta|H} \sum_{i \in [t]} \alpha_t^i \left[ (V_{h+1}^{k_i} - V_{h+1}^*)(s_{h+1}^{k_i}) \right] + \gamma_t,
\end{aligned}
$$

where the second step holds by Fact 1(a) with $g = 1$ and the fact that $V_{h+1}^{k_i}(s) \geq V_{h+1}^*(s)$ and by noticing that $\alpha_t^0, \sum_{i \in [t]} \alpha_t^i \in \{0,1\}$ with $\alpha_t^0 + \sum_{i \in [t]} \alpha_t^i = 1$ by Fact 3(d) (so that $q_1 \geq q_2$), the third step holds since by definition $q_1^+ \geq q_1$ and by Fact 4 $q_2' \leq q_2$, and the last step holds by Fact 1(b) and the fact that $H \geq r_h(s,a) + V_{h+1}^{k_i}(s) \geq r_h(s,a) + V_{h+1}^*(s) \geq 0$. For $G_2$, we have

$$
\begin{aligned}
G_2 &= \frac{1}{\beta} \log\{q_2\} - \frac{1}{\beta} \log\{q_3\} \\
&\leq \frac{1}{\beta}(q_2 - q_3) \\
&\leq \frac{1}{\beta}(q_2^+ - q_3) \\
&= \frac{\alpha_t^0}{\beta} \left[ e^{\beta H} - e^{\beta \cdot Q_h^*(s,a)} \right] + \frac{1}{\beta} \sum_{i \in [t]} \alpha_t^i b_i \\
&\quad + \frac{1}{\beta} \sum_{i \in [t]} \alpha_t^i \left[ e^{\beta[r_h(s,a) + V_{h+1}^*(s_{h+1}^{k_i})]} - \mathbb{E}_{s' \sim P_h(\cdot \mid s,a)} e^{\beta[r_h(s,a) + V_{h+1}^*(s')]} \right] \\
&\leq \alpha_t^0 H e^{|\beta|H} + \gamma_t,
\end{aligned}
$$

In the above, the second step holds by Fact 1(a) with $g = 1$ and

$$\sum_{i \in [t]} \alpha_t^i b_i \geq \left| \sum_{i \in [t]} \alpha_t^i \left[ e^{\beta[r_h(s,a) + V_{h+1}^*(s_{h+1}^{k_i})]} - \mathbb{E}_{s' \sim P_h(\cdot \mid s,a)} e^{\beta[r_h(s,a) + V_{h+1}^*(s')]} \right] \right|$$

on the event of Lemma 9 (so that $q_2 \geq q_3$); the third step holds by Fact 4; the last step holds by Fact 1(b) and $Q_h^*(s,a) \in [0, H]$ and on the event of Lemma 9.

**Case $\beta < 0$.** We have

$$
\begin{aligned}
G_1 &= \frac{1}{(-\beta)} \log\{q_2\} - \frac{1}{(-\beta)} \log\{q_1\} \\
&\leq \frac{e^{-\beta H}}{(-\beta)} (q_2 - q_1) \\
&\leq \frac{e^{-\beta H}}{(-\beta)} (q_2' - q_1^+) \\
&= \frac{e^{-\beta H}}{(-\beta)} \sum_{i \in [t]} \alpha_t^i \left[ e^{\beta [r_h(s,a) + V_{h+1}^*(s_{h+1}^{k_i})]} - e^{\beta [r_h(s,a) + V_{h+1}^{k_i}(s_{h+1}^{k_i})]} \right] + \frac{e^{-\beta H}}{(-\beta)} \sum_{i \in [t]} \alpha_t^i b_i \\
&\leq e^{|\beta| H} \sum_{i \in [t]} \alpha_t^i \left[ (V_{h+1}^{k_i} - V_{h+1}^*)(s_{h+1}^{k_i}) \right] + \gamma_t,
\end{aligned}
$$

where the second step holds by Fact 1(a) with $g = e^{\beta H}$ and the fact that $V_{h+1}^{k_i}(s) \geq V_{h+1}^*(s)$ (so that $q_2 \geq q_1$), the third step holds since $q_2' \geq q_2$ by Fact 4 and $q_1^+ \leq q_1$ by definition, and the last step holds by Fact 1(b) and the fact that $H \geq r_h(s,a) + V_{h+1}^{k_i}(s) \geq r_h(s,a) + V_{h+1}^*(s) \geq 0$. For $G_2$, we have

$$
\begin{aligned}
G_2 &= \frac{1}{(-\beta)} \log\{q_3\} - \frac{1}{(-\beta)} \log\{q_2\} \\
&\leq \frac{e^{-\beta H}}{(-\beta)} (q_3 - q_2) \\
&\leq \frac{e^{-\beta H}}{(-\beta)} (q_3 - q_2^+) \\
&= \frac{e^{-\beta H}}{(-\beta)} \alpha_t^0 \left[ e^{\beta \cdot Q_h^*(s,a)} - e^{\beta H} \right] + \frac{e^{-\beta H}}{(-\beta)} \sum_{i \in [t]} \alpha_t^i b_i \\
&\quad + \frac{e^{-\beta H}}{(-\beta)} \sum_{i \in [t]} \alpha_t^i \left[ \mathbb{E}_{s' \sim P_h(\cdot \mid s,a)} e^{\beta [r_h(s,a) + V_{h+1}^*(s')]} - e^{\beta [r_h(s,a) + V_{h+1}^*(s_{h+1}^{k_i})]} \right] \\
&\leq e^{-\beta H} \alpha_t^0 [H - Q_h^*(s,a)] + \frac{2e^{-\beta H}}{(-\beta)} \sum_{i \in [t]} \alpha_t^i b_i \\
&\leq \alpha_t^0 H e^{|\beta| H} + \gamma_t.
\end{aligned}
$$

where the second step holds by Fact 1(a) given $q_3 \geq q_2$, the second to the last step holds by Fact 1(b), the fact that $Q_h^*(s,a) \leq H$ and on the event of Lemma 9, and the last step holds by the definition of $\gamma_t$.

Combining the bounds of $G_1$ and $G_2$ with the identity $(Q_h^k - Q_h^*)(s,a) = G_1 + G_2$ yields the upper bound for $(Q_h^k - Q_h^*)(s,a)$. The proof is completed in view of Lemma 9 and the definition of $\gamma_t$ that imply

$$
\gamma_t \leq \frac{4c(e^{|\beta| H} - 1)}{|\beta|} \sqrt{\frac{H\iota}{t}}.
$$

$\square$

# E  Proof of Theorem 2

We first introduce some notations. Let $\mathcal{G}$ be a discrete space. Define the shorthand

$$
\mathsf{lse}_\beta(P, f) := \frac{1}{\beta} \log \{ \mathbb{E}_{x \sim P} [\exp(\beta \cdot f(x))] \}, \tag{28}
$$

for a probability distribution $P$ supported on $\mathcal{G}$ and function $f : \mathcal{G} \to \mathbb{R}$. We record a useful lemma that shows $\mathsf{lse}_\beta(\cdot, \cdot)$ is Lipschitz continuous in the second argument.

**Lemma 12.** *Let $\mathcal{G}$ be a discrete space and $\bar{f} \geq 0$ be a non-negative number. Let the functions $f, f' : \mathbb{R}^d \mapsto [0, \bar{f}]$ be such that $f(x) \geq f'(x)$ for all $x \in \mathbb{R}^d$. Also let $P$ be a probability distribution supported on $\mathcal{G}$. We have*

$$\mathsf{lse}_\beta(P, f) - \mathsf{lse}_\beta(P, f') \leq e^{|\beta|\bar{f}} \cdot \mathbb{E}_{x \sim P}[f(x) - f'(x)].$$

The proof is given in Appendix E.1.

Define $\hat{P}_h^k(\cdot \mid s, a)$ to be the delta function centered at $s_{h+1}^k$ for all $(k, h, s, a) \in [K] \times [H] \times \mathcal{S} \times \mathcal{A}$, and this means $\mathbb{E}_{s' \sim \hat{P}_h^k(\cdot \mid s, a)}[f(s')] = f(s_{h+1}^k)$ for any function $f : \mathcal{S} \to \mathbb{R}$. We let

$$\delta_h^k := (V_h^k - V_h^{\pi_k})(s_h^k) \quad \text{and} \quad \phi_h^k := (V_h^k - V_h^*)(s_h^k).$$

Also define

$$\xi_{h+1}^k := [(P_h - \hat{P}_h^k)(V_{h+1}^* - V_{h+1}^{\pi_k})](s_h^k, a_h^k).$$

Note that For each $(k, h) \in [K] \times [H]$, we have

$$
\begin{aligned}
\delta_h^k &= (Q_h^k - Q_h^{\pi_k})(s_h^k, a_h^k) \\
&= (Q_h^k - Q_h^*)(s_h^k, a_h^k) + (Q_h^* - Q_h^{\pi_k})(s_h^k, a_h^k) \\
&\leq \alpha_t^0 H e^{|\beta|H} + \sum_{i \in [t]} \alpha_t^i e^{|\beta|H} \phi_{h+1}^{k_i} + 2\gamma_t \\
&\quad + [\mathsf{lse}(P_h(\cdot \mid s_h^k, a_h^k), V_{h+1}^*) - \mathsf{lse}(P_h(\cdot \mid s_h^k, a_h^k), V_{h+1}^{\pi_k})] \\
&\leq \alpha_t^0 H e^{|\beta|H} + \sum_{i \in [t]} \alpha_t^i e^{|\beta|H} \phi_{h+1}^{k_i} + 2\gamma_t + e^{|\beta|H}[P_h(V_{h+1}^* - V_{h+1}^{\pi_k})](s_h^k, a_h^k) \\
&= \alpha_t^0 H e^{|\beta|H} + \sum_{i \in [t]} \alpha_t^i e^{|\beta|H} \phi_{h+1}^{k_i} + 2\gamma_t + e^{|\beta|H}(\delta_{h+1}^k - \phi_{h+1}^k + \xi_{h+1}^k), \qquad (29)
\end{aligned}
$$

where the third step holds by Lemma 11 and the Bellman equations (3) and (4), the fourth step holds by Lemma 12 and the fact that $0 \leq V_{h+1}^{\pi_k}(s) \leq V_{h+1}^*(s) \leq H$ for all $s \in \mathcal{S}$, and the last step follows by defintion that $\delta_{h+1}^k - \phi_{h+1}^k = (V_{h+1}^* - V_{h+1}^{\pi_k})(s_{h+1}^k) = [\hat{P}_h^k(V_{h+1}^* - V_{h+1}^{\pi_k})](s_h^k, a_h^k)$ and the definition of $\xi_{h+1}^k$.

We now compute $\sum_{k \in [K]} \delta_h^k$ for a fixed $h \in [H]$. Denote by $n_h^k := N_h^k(s_h^k, a_h^k)$ and we have

$$\sum_{k \in [K]} \alpha_{n_h^k}^0 H e^{|\beta|H} = H e^{|\beta|H} \sum_{k \in [K]} \mathbb{I}\{n_h^k = 0\} \leq H e^{|\beta|H} SA.$$

Then we turn to control the second term in Equation (29) summed over $k \in [K]$, that is,

$$\sum_{k \in [K]} \sum_{i \in [t]} \alpha_t^i e^{|\beta|H} \phi_{h+1}^{k_i} = e^{|\beta|H} \sum_{k \in [K]} \sum_{i \in [n_h^k]} \alpha_{n_h^k}^i \phi_{h+1}^{k_i(s_h^k, a_h^k)},$$

where $k_i(s_h^k, a_h^k)$ denotes the episode in which $(s_h^k, a_h^k)$ was taken at step $h$ for the $i$-th time. We re-group the above summation in a different way. For every $k' \in [K]$, the term $\phi_{h+1}^{k'}$ appears in the summand with $k > k'$ if and only if $(s_h^k, a_h^k) = (s_h^{k'}, a_h^{k'})$. The first time it appears we have $n_h^k = n_h^{k'} + 1$, the second time it appears we have $n_h^k = n_h^{k'} + 2$, and etc. Therefore,

$$e^{|\beta|H} \sum_{k \in [K]} \sum_{i \in [n_h^k]} \alpha_{n_h^k}^i \phi_{h+1}^{k_i(s_h^k, a_h^k)} \leq e^{|\beta|H} \sum_{k' \in [K]} \phi_{h+1}^{k'} \sum_{t \geq n_h^{k'} + 1} \alpha_t^{n_h^{k'}} \leq e^{|\beta|H} \left(1 + \frac{1}{H}\right) \sum_{k' \in [K]} \phi_{h+1}^{k'},$$

where the last step follows Fact 3(c). Collecting the above results and plugging them into Equation (29), we have

$$\sum_{k \in [K]} \delta_h^k \leq H e^{|\beta|H} SA + e^{|\beta|H} \left(1 + \frac{1}{H}\right) \sum_{k \in [K]} \phi_{h+1}^k$$

$$+ e^{|\beta|H} \sum_{k\in[K]} (\delta_{h+1}^k - \phi_{h+1}^k) + \sum_{k\in[K]} (2\gamma_{n_h^k} + e^{|\beta|H}\xi_{h+1}^k)$$

$$\le He^{|\beta|H}SA + e^{|\beta|H}\left(1 + \frac{1}{H}\right)\sum_{k\in[K]}\delta_{h+1}^k$$

$$+ \sum_{k\in[K]} (2\gamma_{n_h^k} + e^{|\beta|H}\xi_{h+1}^k), \tag{30}$$

where the last step holds since $\delta_{h+1}^k \ge \phi_{h+1}^k$ (due to the fact that $V_{h+1}^*(s) \ge V_{h+1}^{\pi_k}(s)$ for all $x \in \mathcal{S}$). Since it holds that

$$\left[e^{|\beta|H}\left(1 + \frac{1}{H}\right)\right]^H \le e^{|\beta|H^2+1},$$

we can expand the quantity $\sum_{k\in[K]}\delta_1^k$ recursively in the form of Equation (30), apply Holder's inequality and use the fact that $\delta_{H+1}^k = 0$ to get

$$\sum_{k\in[K]}\delta_1^k \le e^{|\beta|H^2+1}\left[H^2 e^{|\beta|H}SA + \sum_{h\in[H]}\sum_{k\in[K]}(2\gamma_{n_h^k} + e^{|\beta|H}\xi_{h+1}^k)\right]. \tag{31}$$

By the pigeonhole principle, for any $h \in [H]$ we have

$$\sum_{k\in[K]}\gamma_{n_h^k} \lesssim \frac{e^{|\beta|H}-1}{|\beta|}\sum_{k\in[K]}\sqrt{\frac{H\iota}{n_h^k}}$$

$$= \frac{e^{|\beta|H}-1}{|\beta|}\sum_{(s,a)\in\mathcal{S}\times\mathcal{A}}\sum_{n\in[N_h^K(s,a)]}\sqrt{\frac{H\iota}{n}}$$

$$\lesssim \frac{e^{|\beta|H}-1}{|\beta|}\sqrt{HSAK\iota}$$

$$= \frac{e^{|\beta|H}-1}{|\beta|}\sqrt{SAT\iota}, \tag{32}$$

where the third step holds since $\sum_{(s,a)\in\mathcal{S}\times\mathcal{A}} N_h^K(s,a) = K$ and the RHS of the second step is maximized when $N_h^K(s,a) = K/(SA)$ for all $(s,a) \in \mathcal{S}\times\mathcal{A}$. Finally, the Azuma-Hoeffding inequality implies that with probability at least $1 - \delta$, we have

$$\left|\sum_{h\in[H]}\sum_{k\in[K]}\xi_{h+1}^k\right| \lesssim H\sqrt{T\iota}. \tag{33}$$

Putting together Equations (32) and (33) and plugging them into (31), we have

$$\sum_{k\in[K]}\delta_1^k \lesssim e^{|\beta|(H^2+H)} \cdot H^2 SA$$

$$+ e^{|\beta|H^2} \cdot \frac{e^{|\beta|H}-1}{|\beta|}\sqrt{H^2 SAT\iota}$$

$$+ e^{|\beta|(H^2+H)} \cdot H\sqrt{T\iota}.$$

$$\le e^{|\beta|(H^2+H)} \cdot H^2 SA$$

$$+ e^{|\beta|(H^2+H)} \cdot \frac{e^{|\beta|H}-1}{|\beta|}\sqrt{H^2 SAT\iota}$$

The proof is completed in view of Fact 2 and when $T$ is sufficiently large.

### E.1 Proof of Lemma 12

We have the following two cases.

**Case $\beta > 0$.** We have

$$
\begin{aligned}
\mathsf{lse}_\beta(P, f) - \mathsf{lse}_\beta(P, f') &\leq \frac{1}{\beta}\mathbb{E}_{x\sim P}\left[e^{\beta \cdot f(x)} - e^{\beta \cdot f'(x)}\right] \\
&\leq \frac{1}{\beta}\mathbb{E}_{x\sim P}\left[\beta e^{\beta \bar{f}}(f(x) - f'(x))\right] \\
&= e^{\beta \bar{f}} \cdot \mathbb{E}_{x\sim P}[f(x) - f'(x)],
\end{aligned}
$$

where the first step holds by Fact 1(a) with $g = 1$ and the fact that $e^{\beta \cdot f(x)} \geq e^{\beta \cdot f'(x)} \geq 1$, and the second holds by Fact 1(b) with $u = \bar{f}$ and the fact that $f(x) \geq f'(x)$.

**Case $\beta < 0$.** We have

$$
\begin{aligned}
\mathsf{lse}_\beta(P, f) - \mathsf{lse}_\beta(P, f') &= -\left[\mathsf{lse}_\beta(P, f') - \mathsf{lse}_\beta(P, f)\right] \\
&\leq \frac{\exp(-\beta\bar{f})}{(-\beta)}\mathbb{E}_{x\sim P}\left[\exp(\beta \cdot f'(x)) - \exp(\beta \cdot f(x))\right] \\
&\leq \frac{\exp(-\beta\bar{f})}{(-\beta)}\mathbb{E}_{x\sim P}\left[(-\beta)(f(x) - f'(x))\right] \\
&= \exp(-\beta\bar{f}) \cdot \mathbb{E}_{x\sim P}[f(x) - f'(x)],
\end{aligned}
$$

where the second step holds by Fact 1(a) with $g = e^{\beta\bar{f}}$ given that $x \in [e^{\beta\bar{f}}, 1]$, and the third step holds by Fact 1(b) and the fact $1 \geq e^{\beta \cdot f'(x)} \geq e^{\beta \cdot f(x)} > 0$.

## F Proof of Theorem 3

We consider the following MDP as illustrated in Figure 2. For now, we focus on the case $\beta > 0$; we shall see soon that the construction for $\beta < 0$ can be done in a similar way. The MDP is equipped with $\mathcal{A} = \{a_1, a_2\}$ and $\mathcal{S} = \{s_1, s_2, s_3\}$, where state $s_1$ is the initial state, and states $s_2$ and $s_3$ are absorbing regardless of actions taken. The reward function satisfies that $r_h(s_2, a) = 1$ and $r_h(s_1, a) = r_h(s_3, a) = 0$ for all $h \in [H]$ and $a \in \mathcal{A}$. In Figure 2, step $H + 1$ is a virtual step that represents termination of an episode and generates no reward. At the initial state $s_1$, we may choose to take action $a_1$ or $a_2$. If $a_1$ is taken at state $s_1$, then we transition to $s_2$ with probability $p_1$ and to $s_3$ with probability $1 - p_1$. If $a_2$ is taken at state $s_1$, then we transition to $s_2$ with probability $p_2$ and to $s_3$ with probability $1 - p_2$. We interact with such an MDP for $K$ episodes.

We note that the above $K$-episode MDP is equivalent to a $K$-round two-armed bandit with per-round reward ranging in $[0, H - 1]$, where the first transition in each episode of the MDP can be viewed as a pull of an arm in each round of the bandit. Therefore, the regret lower bound for the MDP can be proved using lower bound techniques for bandits. Our proof follows the same reasoning of [41, Theorem 15.2]. We start by discussing the setup for the proof under the cases $\beta > 0$ and $\beta < 0$.

For each $\rho \in [0, 1]$, let $\mathrm{Ber}(\rho)$ denote the Bernoulli distribution with parameter $\rho$. Let us fix a policy $\pi$. We first consider the case $\beta > 0$. We construct a pair of two-armed bandits, which we call $\nu_p$ and $\nu_{p'}$. The first bandit $\nu_p$ has $X_1 = (H - 1) \cdot \mathrm{Ber}(p_1)$ as the first arm and $X_2 = (H - 1) \cdot \mathrm{Ber}(p_2)$ as the second arm. The second bandit $\nu_{p'}$ has $X_1' = (H - 1) \cdot \mathrm{Ber}(p_1')$ as the first arm and $X_2' = (H - 1) \cdot \mathrm{Ber}(p_2')$ as the second arm. We let $p_2 < p_1 = p_1' < p_2'$ and $p_2 = e^{-\beta(H-1)}$. Let $\Delta := p_1 - p_2$ and we will choose $\Delta \leq \frac{1}{4}e^{-\beta(H-1)}$ later in the proof. Note that when $|\beta(H - 1)|$ is large enough, we have $\Delta \leq \frac{1}{100}$. Let $p_2' = p_1 + \Delta$, so that $p_2' = p_2 + 2\Delta$ and $p_2' \leq \frac{1}{4}$.

We then consider $\beta < 0$. Let $p_2 = e^{\beta(H-1)}$, and set $p_1 = p_1' = p_2 - \Delta$ and $p_2' = p_2 - 2\Delta$ for some $\Delta \in (0, \frac{1}{4}e^{\beta(H-1)}]$ to be specified later. Similar to the case $\beta > 0$, we construct a pair of two-armed bandits $\nu_p$ and $\nu_{p'}$. The first bandit $\nu_p$ has $X_1 = (H-1) \cdot \mathrm{Ber}(1 - p_1)$ as the first arm and $X_2 = (H-1) \cdot \mathrm{Ber}(1 - p_2)$ as the second arm. The second bandit $\nu_{p'}$ has $X_1' = (H-1) \cdot \mathrm{Ber}(1 - p_1')$ as the first arm and $X_2' = (H-1) \cdot \mathrm{Ber}(1 - p_2')$ as the second arm. When $|\beta(H-1)|$ is sufficiently

Figure 2: Illustration of the MDP for the lower bound proof for $\beta > 0$.

large, we have $1 - p_2' \geq 1 - p_1' = 1 - p_1 \geq 1 - p_2 = 1 - e^{\beta(H-1)} \geq \frac{1}{2}$ and $\Delta \leq \frac{1}{4}e^{\beta(H-1)}$ implies $\Delta \leq \frac{1}{100}$.

In the remaining of the section, we provide a unified proof for both cases of $\beta > 0$ and $\beta < 0$. We denote by $\mathbb{P}_{\pi,\nu_p}$ and $\mathbb{P}_{\pi,\nu_{p'}}$ the probability measures induced jointly by $\pi$ and the two bandits, respectively. We will use the shorthands $\mathbb{P}_p := \mathbb{P}_{\pi,\nu_p}$ and $\mathbb{P}_{p'} := \mathbb{P}_{\pi,\nu_{p'}}$ for notational simplicity. Note that for both $\beta > 0$ and $\beta < 0$, the first arm is optimal for bandit $\nu_p$, while the second is optimal for bandit $\nu_{p'}$. Let $T_a(K)$ be the number of times we have pulled the $a$-th arm of a bandit after we execute policy $\pi$ for $K$ rounds. It is clear that $\mathbb{E}_p[T_2(K)] \leq K$. Let $R_{\pi,\nu}(K)$ denote the regret of policy $\pi$ after it is executed for $K$ rounds in bandit $\nu$.

By Lemmas 13 and 14, we have

$$
\begin{aligned}
R_{\pi,\nu_p}(K) &\gtrsim \frac{e^{|\beta|(H-1)} - 1}{|\beta|} \cdot \Delta \cdot \mathbb{E}_p[T_2(K)] \\
&\geq \frac{e^{|\beta|(H-1)} - 1}{|\beta|} \cdot \Delta \cdot \left[ \frac{K}{2} \cdot \mathbb{P}_p(T_1(K) \leq K/2) \right],
\end{aligned}
$$

and

$$
\begin{aligned}
R_{\pi,\nu_{p'}}(K) &\gtrsim \frac{e^{|\beta|(H-1)} - 1}{|\beta|} \cdot \Delta \cdot \mathbb{E}_{p'}[T_1(K)] \\
&\geq \frac{e^{|\beta|(H-1)} - 1}{|\beta|} \cdot \Delta \cdot \left[ \frac{K}{2} \cdot \mathbb{P}_{p'}(T_1(K) > K/2) \right].
\end{aligned}
$$

We combine the above two displays and get

$$
\begin{aligned}
&\frac{1}{2}\left[ R_{\pi,\nu_p}(K) + R_{\pi,\nu_{p'}}(K) \right] \\
&\gtrsim \frac{e^{|\beta|(H-1)} - 1}{|\beta|} \cdot K\Delta \left[ \mathbb{P}_p(T_1(K) \leq K/2) + \mathbb{P}_{p'}(T_1(K) > K/2) \right] \\
&\geq \frac{e^{|\beta|(H-1)} - 1}{|\beta|} \cdot K\Delta \cdot \exp\left[ -D_{\mathrm{KL}}(\mathbb{P}_p \| \mathbb{P}_{p'}) \right] \\
&\geq \frac{e^{|\beta|(H-1)} - 1}{|\beta|} \cdot K\Delta \cdot \exp\left[ -K \cdot \frac{8\Delta^2}{p_2(1-p_2)} \right]
\end{aligned}
$$

(34)

where the second step holds by the Bretagnolle–Huber inequality [41, Theorem 14.2], and the last step follows from the fact that $\mathbb{E}_p[T_2(K)] \leq K$ and Lemma 16. Now we set

$$
\Delta := \sqrt{\frac{p_2(1-p_2)}{K}}.
$$

Note that this choice of $\Delta$ ensures $\Delta \leq \frac{1}{4}e^{-|\beta|(H-1)}$ as long as $K$ is sufficiently large. Hence, continuing from (34) we have

$$
\begin{aligned}
\frac{1}{2}\left[R_{\pi,\nu_p}(K) + R_{\pi,\nu_{p'}}(K)\right] &\gtrsim \frac{e^{|\beta|(H-1)} - 1}{|\beta|} \cdot K\Delta \\
&\gtrsim \frac{e^{|\beta|(H-1)} - 1}{|\beta|} \cdot \sqrt{p_2(1 - p_2)K} \\
&\geq \frac{e^{|\beta|(H-1)} - 1}{|\beta|} \cdot \sqrt{\frac{1}{2}e^{-\beta(H-1)}K} \\
&\gtrsim \frac{e^{|\beta|(H-1)/2} - 1}{|\beta|}\sqrt{K},
\end{aligned}
$$

where the third step holds since $p_2 = e^{-|\beta|(H-1)}$ and $1 - p_2 \geq \frac{1}{2}$. The proof is completed by upper bounding the LHS of the above display by $\max\{R_{\pi,\nu_p}(K), R_{\pi,\nu_{p'}}(K)\}$, and recalling that $\lambda(u) = (e^{3u} - 1)/u$ for $u > 0$ and $T = KH$.

### F.1 Auxiliary Lemmas

**Lemma 13.** *Let $\pi$ be any policy and $\nu$ be any two-armed bandit with distinct arms. Let $X_a$ denote the $a$-th arm of $\nu$. Define $a^* := \text{argmax}_{a\in\{a_1,a_2\}} \frac{1}{\beta}\log \mathbb{E}_\nu e^{\beta X_a}$ and $b \in \{a_1, a_2\}\backslash\{a^*\}$. Also define*

$$
\delta_{b,\nu} := \begin{cases} (\mathbb{E}_\nu e^{\beta X_{a^*}} - \mathbb{E}_\nu e^{\beta X_b})/\mathbb{E}_\nu e^{\beta X_{a^*}}, & \text{if } \beta > 0, \\ (\mathbb{E}_\nu e^{\beta X_b} - \mathbb{E}_\nu e^{\beta X_{a^*}})/\mathbb{E}_\nu e^{\beta X_{a^*}}, & \text{if } \beta < 0. \end{cases}
$$

*We have*

$$
R_{\pi,\nu}(K) \geq \frac{1}{2|\beta|}\delta_{b,\nu} \cdot \mathbb{E}_{\pi,\nu}\left[T_b(K)\right].
$$

*Proof.* Let $Y_k$ be the reward received at round $k$ by executing $\pi$, and $\mathcal{A} = \{a^*, b\}$. We slightly abuse the notation by writing $\pi^k = a$ to mean that arm $a$ is pulled in round $k$ by executing $\pi$. Recall the definitions of the value functions $V_1^*$ and $V_1^\pi$ from (4) and (3), respectively. Since there is no state in bandit, we omit the arguments of the value functions. We observe that

$$
V_1^* = \frac{1}{\beta}\log \mathbb{E}e^{\beta X_{a^*}}
$$

and

$$
V_1^{\pi^k} = \frac{1}{\beta}\log \mathbb{E}e^{\beta Y_k} = \frac{1}{\beta}\log\left\{\sum_{a\in\mathcal{A}}\mathbb{P}(\pi^k = a)\cdot \mathbb{E}e^{\beta X_a}\right\}.
$$

In the RHS of the two displays above, the probability $\mathbb{P}(\cdot)$ is with respect to $\pi$ and $\nu$, and the expectation $\mathbb{E}[\cdot]$ is with respect to $\nu$. Note that by the definitions of $a^*$ and $b$, we have $\delta_{b,\nu} \in [0, 1]$ for any $\beta \neq 0$.

For $\beta > 0$, we have

$$
\begin{aligned}
V_1^* - V_1^{\pi^k} &= \frac{1}{\beta}\log\left\{\frac{\sum_{a\in\mathcal{A}}\mathbb{P}(\pi^k = a)\cdot \mathbb{E}e^{\beta X_{a^*}}}{\sum_{a\in\mathcal{A}}\mathbb{P}(\pi^k = a)\cdot \mathbb{E}e^{\beta X_a}}\right\} \\
&= \frac{1}{\beta}\log\left\{1 + \frac{\mathbb{P}(\pi^k = b)\cdot(\mathbb{E}e^{\beta X_{a^*}} - \mathbb{E}e^{\beta X_b})}{\sum_{a\in\mathcal{A}}\mathbb{P}(\pi^k = a)\cdot \mathbb{E}e^{\beta X_a}}\right\} \\
&\geq \frac{1}{\beta}\log\left\{1 + \frac{\mathbb{P}(\pi^k = b)\cdot(\mathbb{E}e^{\beta X_{a^*}} - \mathbb{E}e^{\beta X_b})}{\mathbb{E}e^{\beta X_{a^*}}}\right\} \\
&= \frac{1}{\beta}\log\left\{1 + \mathbb{E}\left[\mathbb{I}\{\pi^k = b\}\right]\cdot \delta_{b,\nu}\right\} \\
&\geq \frac{1}{2\beta}\cdot \mathbb{E}\left[\mathbb{I}\{\pi^k = b\}\right]\cdot \delta_{b,\nu},
\end{aligned}
$$

where the last step holds since $\delta_{b,\nu} \in [0,1]$ and $\log(1+x) \geq \frac{x}{2}$ for $x \in [0,1]$. Summing both sides of the above display over $k \in [K]$ and noticing $T_b(K) = \sum_{k \in [K]} \mathbb{I}\{\pi^k = b\}$ yield the result.

For $\beta < 0$, we have

$$
\begin{aligned}
V_1^* - V_1^{\pi^k} &= \frac{1}{|\beta|} \log \left\{ \sum_{a \in \mathcal{A}} \mathbb{P}(\pi^k = a) \cdot \mathbb{E}e^{\beta X_a} \right\} - \frac{1}{|\beta|} \log \mathbb{E}e^{\beta X_{a^*}} \\
&= \frac{1}{|\beta|} \log \left\{ \frac{\sum_{a \in \mathcal{A}} \mathbb{P}(\pi^k = a) \cdot \mathbb{E}e^{\beta X_a}}{\mathbb{E}e^{\beta X_{a^*}}} \right\} \\
&= \frac{1}{|\beta|} \log \left\{ 1 + \frac{\mathbb{P}(\pi^k = b) \cdot (\mathbb{E}e^{\beta X_b} - \mathbb{E}e^{\beta X_{a^*}})}{\mathbb{E}e^{\beta X_{a^*}}} \right\} \\
&= \frac{1}{|\beta|} \log \left\{ 1 + \mathbb{E}\left[\mathbb{I}\{\pi^k = b\}\right] \cdot \delta_{b,\nu} \right\} \\
&\geq \frac{1}{2|\beta|} \cdot \mathbb{E}\left[\mathbb{I}\{\pi^k = b\}\right] \cdot \delta_{b,\nu},
\end{aligned}
$$

where the last step holds since $\delta_{b,\nu} \in [0,1]$ and $\log(1+x) \geq \frac{x}{2}$ for $x \in [0,1]$. Summing both sides of the above display over $k \in [K]$ and noticing $T_b(K) = \sum_{k \in [K]} \mathbb{I}\{\pi^k = b\}$ yield the result. $\qquad \square$

**Lemma 14.** *Consider the setting of Lemma 13, and recall the bandits $\nu_p$ and $\nu_{p'}$ and the quantity $\Delta$ defined in Section F. For $\nu \in \{\nu_p, \nu_{p'}\}$, we have*

$$
\delta_{b,\nu} \gtrsim \Delta(e^{|\beta|(H-1)} - 1).
$$

*Proof.* We first consider the case $\beta > 0$. For $\nu = \nu_p$, we have

$$
\begin{aligned}
\delta_{b,\nu} &= \frac{p_1 e^{\beta(H-1)} + (1 - p_1) - [p_2 e^{\beta(H-1)} + (1 - p_2)]}{p_1 e^{\beta(H-1)} + (1 - p_1)} \\
&= \frac{\Delta(e^{\beta(H-1)} - 1)}{p_1 e^{\beta(H-1)} + (1 - p_1)} \\
&\geq \frac{\Delta(e^{\beta(H-1)} - 1)}{3},
\end{aligned}
$$

where the second step holds since $p_1 = p_2 + \Delta$, and the last step holds since $p_1 = p_2 + \Delta \leq 2e^{-\beta(H-1)}$ given $p_2 = e^{-\beta(H-1)}$ and $\Delta \leq \frac{1}{4}e^{-\beta(H-1)}$. For $\nu = \nu_{p'}$, we have

$$
\begin{aligned}
\delta_{b,\nu} &= \frac{p_2' e^{\beta(H-1)} + (1 - p_2') - [p_1' e^{\beta(H-1)} + (1 - p_1')]}{p_2' e^{\beta(H-1)} + (1 - p_2')} \\
&= \frac{\Delta(e^{\beta(H-1)} - 1)}{p_2' e^{\beta(H-1)} + (1 - p_2')} \\
&\geq \frac{\Delta(e^{\beta(H-1)} - 1)}{4},
\end{aligned}
$$

where the second step holds since $p_2' = p_1' + \Delta = p_1 + \Delta$, and the last step holds since $p_2' = p_1 + \Delta = p_2 + 2\Delta \leq 3e^{-\beta(H-1)}$ given $p_2 = e^{-\beta(H-1)}$ and $\Delta \leq e^{-\beta(H-1)}$.

Now we consider $\beta < 0$. For $\nu = \nu_p$, we have

$$
\begin{aligned}
\delta_{b,\nu} &= \frac{(1 - p_2)e^{\beta(H-1)} + p_2 - [(1 - p_1)e^{\beta(H-1)} + p_1]}{(1 - p_1)e^{\beta(H-1)} + p_1} \\
&= \frac{\Delta(1 - e^{\beta(H-1)})}{(1 - p_1)e^{\beta(H-1)} + p_1} \\
&\geq \frac{\Delta(1 - e^{\beta(H-1)})}{2e^{\beta(H-1)}}
\end{aligned}
$$

$$= \frac{\Delta(e^{-\beta(H-1)} - 1)}{2},$$

where the second step holds since $p_1 = p_2 - \Delta$, and the third step holds since $1 - p_1 \leq 1$ and $p_1 = p_2 - \Delta = e^{\beta(H-1)} - \Delta \leq e^{\beta(H-1)}$. For $\nu = \nu_{p'}$, we have

$$\begin{aligned} \delta_{b,\nu} &= \frac{(1 - p_1')e^{\beta(H-1)} + p_1' - [(1 - p_2')e^{\beta(H-1)} + p_2']}{(1 - p_2')e^{\beta(H-1)} + p_2'} \\ &= \frac{\Delta(1 - e^{\beta(H-1)})}{(1 - p_2')e^{\beta(H-1)} + p_2'} \\ &\geq \frac{\Delta(1 - e^{\beta(H-1)})}{2e^{\beta(H-1)}} \\ &= \frac{\Delta(e^{-\beta(H-1)} - 1)}{2}, \end{aligned}$$

where the second step holds since $p_2' = p_1' - \Delta$, and the third step holds since $1 - p_2' \leq 1$ and $p_2' = p_2 - 2\Delta = e^{\beta(H-1)} - 2\Delta \leq e^{\beta(H-1)}$. We note $-\beta(H-1) = |\beta|(H-1)$ since $\beta < 0$ and the proof is completed. $\qquad\square$

**Lemma 15.** *Under the setting of Section F, we have*

$$D_{KL}(\mathbb{P}_p \| \mathbb{P}_{p'}) \leq K \cdot \frac{8\Delta^2}{p_2(1 - p_2)}.$$

*Proof.* For $\beta > 0$, we have

$$\begin{aligned} D_{\mathrm{KL}}(\mathbb{P}_p \| \mathbb{P}_{p'}) &= \mathbb{E}_p\left[T_2(K)\right] \cdot D_{\mathrm{KL}}(\mathrm{Ber}(p_2) \| \mathrm{Ber}(p_2')) \\ &\leq K \cdot \frac{(p_2' - p_2)^2}{p_2'(1 - p_2')} \\ &= K \cdot \frac{4\Delta^2}{p_2'(1 - p_2')} \\ &\leq K \cdot \frac{4\Delta^2}{p_2(1 - p_2)}, \end{aligned}$$

where the first step follows from [41, Lemma 15.1], the second step follows from the fact that $\mathbb{E}_p\left[T_2(K)\right] \leq K$ and Lemma 16, the third step follows from the identity $p_2' = p_2 + 2\Delta$, and the last step holds since $p_2 \leq p_2' \leq \frac{1}{2}$ and the function $x \mapsto x(1 - x)$ is increasing on $[0, \frac{1}{2}]$.

For $\beta < 0$, we have

$$\begin{aligned} D_{\mathrm{KL}}(\mathbb{P}_p \| \mathbb{P}_{p'}) &= \mathbb{E}_p\left[T_2(K)\right] \cdot D_{\mathrm{KL}}(\mathrm{Ber}(1 - p_2) \| \mathrm{Ber}(1 - p_2')) \\ &\leq K \cdot \frac{(p_2' - p_2)^2}{p_2'(1 - p_2')} \\ &= K \cdot \frac{4\Delta^2}{p_2'(1 - p_2')} \\ &\leq K \cdot \frac{8\Delta^2}{p_2(1 - p_2)}, \end{aligned}$$

where the first step follows from [41, Lemma 15.1], the second step follows from the fact that $\mathbb{E}_p\left[T_2(K)\right] \leq K$ and Lemma 16, the third step follows from the identity $p_2' = p_2 - 2\Delta$, and the last step holds since $p_2 = e^{\beta(H-1)}$ and $\Delta \leq \frac{1}{4}e^{\beta(H-1)} = \frac{1}{4}p_2$ means $\frac{1}{2}p_2 \leq p_2' \leq p_2 \leq \frac{1}{2}$ which implies $p_2(1 - p_2) \leq 2p_2'(1 - p_2')$. $\qquad\square$

**Lemma 16.** *Let $q, q'$ be such that $0 \leq q' < q < 1$. We have*

$$D_{KL}(Ber(q') \| Ber(q)) \leq \frac{(q - q')^2}{q(1 - q)}.$$

*Proof.* Let $\Delta_q := q - q'$. The KL divergence can be upper bounded as follows:

$$
\begin{aligned}
D_{\mathrm{KL}}\left(\mathrm{Ber}(q')\|\mathrm{Ber}(q)\right) &= q' \log\left(\frac{q'}{q}\right) + (1 - q') \log\left(\frac{1 - q'}{1 - q}\right) \\
&= q' \log\left(1 + \frac{q' - q}{q}\right) + (1 - q') \log\left(1 + \frac{q - q'}{1 - q}\right) \\
&\overset{(i)}{\leq} q' \cdot \frac{q' - q}{q} + (1 - q') \cdot \frac{q - q'}{1 - q} \\
&= (\Delta_q - q) \cdot \frac{\Delta_q}{q} + (1 - q + \Delta_q) \cdot \frac{\Delta_q}{1 - q} \\
&= \frac{\Delta_q^2}{q} + \frac{\Delta_q^2}{1 - q} \\
&= \frac{\Delta_q^2}{q(1 - q)},
\end{aligned}
$$

where step $(i)$ holds since $\log(1 + x) \leq x$ for all $x > -1$. The proof is completed.

$\square$