[Reviews · NeurIPS 2020]

Review 1

Summary and Contributions: This paper proposes a couple of risk-sensitive reinforcement learning algorithms and analyzes their regret. The results illuminate very interesting dependency of the regret on risk-sensitivity. In particular, the their regret recovers the known regret of risk-neutral case as the risk-sensitivity parameter tends to 0, and their regret shows exponential dependency on the risk-sensitivity parameter for both risk-seeking and risk-averse cases. The paper not only discuss the upper bound on the regret but also the lower bound, where the exponential dependency on the risk-sensitivity parameter also appears.

Strengths: The major strength of the paper is in the new insights regarding how the regret depend on the risk-sensitivity. The authors well explain this point in terms of aleatoric uncertainty and epistemic uncertainty. The regret bounds involving the risk-sensitivity parameter is a first of a kind, and the lower and upper bounds are sufficiently tight to discuss the dependency on the risk-sensitivity parameter. In addition, the proposed algorithms appear to be useful in practice.

Weaknesses: The main part of the proofs of the regret bounds are in the supplementary material, which I have not checked, and there are no numerical support for the main results either. So, I have to believe the stated results, but if I believe them, the results are very significant, novel, and relevant.

Correctness: It is hard to judge the correctness. I have not checked the proofs, and there are no numerical support.

Clarity: Very well written.

Relation to Prior Work: Very clear.

Reproducibility: Yes

Additional Feedback: The discussion about scaling around Lemma 1 could also be made with the properties of entropic risk measure such as (1/beta) log E[exp(beta c)] = c for constant c (e.g. footnote 4 of [48]). Step 10 of Algorithm 1 should have dependency on \phi(s,a). I would also like to see discussion about what happens when we use non-linear functional approximator in Algorithm 1. How much does the regret rely on the linear functional approximator in (8)? L218: "bonu as" -- Thank you for the responses.


Review 2

Summary and Contributions: The authors propose two model-free algorithms for reinforcement learning in finite, episodic MDPs with risk-sensitive objective. Regret bounds are derived for the proposed algorithms and are shown to be consistent with existing results under the risk-neutral setting.

Strengths: The results are mainly theoretical. Both the algorithms and the regret bounds seem novel for this setting. The regret bounds also provide useful insights regarding the tradeoff between risk sensitivity and sample efficiency.

Weaknesses: It is unclear whether the proposed algorithms and the regret analysis here have any practical relevance. It is unclear how the universal constant in the exploration bonus should be chosen if one actually wants to implement the algorithms.

Correctness: I did not check the proofs but the results seem plausible.

Clarity: The paper is easy to read. The presentation is clear.

Relation to Prior Work: The authors did a decent job connecting this work with prior works.

Reproducibility: Yes

Additional Feedback: Some empirical demonstrations of the practical relevance of the proposed algorithms would be helpful. ===== Post-Rebuttal ===== I have read the authors' feedback, I maintain my score. Thank you.


Review 3

Summary and Contributions: This paper studies a risk-sensitive RL problem where the value function is log of expectation of exponential with the power being a (scaled) return. The problem statement is well presented and risk-sensitiveness is well explained in this formulation. The paper introduce (is it your contribution?) the Bellman equation for this problem, which is nonlinear in contrast to the risk-neutral (common) RL setting.

Strengths: The paper then proposes two algorithms, one for batch setting, the other for online setting, how to learn the value function using Q-learning, as well as regret bound for the two algorithms.

Weaknesses: The algorithms are actually straightforward and I don't see challenges brought by the nonlinearity (please clarify why nonlinearity is a challenge, what kinds challenge you mean).

Correctness: yes.

Clarity: yes.

Relation to Prior Work: yes.

Reproducibility: Yes

Additional Feedback: The regret in Lemma 1 is also straightforward. The algorithmic idea is to minimizing the Bellman error. (eqn 8), with the TD target being the corresponding exponentialized one. So algorithm novelty is Okay. The key contribution of the paper is the regret bound analysis of the two algorithms, which is mostly built on Taylor series approximation of the value function. Theorem 1 and 2 present the regret bound of the two algorithms, following an analysis similar to UCB, drawing from a Taylor expansion approximation of the (nonlinear) Bellman equation. Theorem 3 shows the lower bound of the algorithms. the b_h was not introduced in algorithm presentation (below eqn 8). It is a UCB term. No experiments were presented (a bit odd for an algorithmic paper). Post-rebuttal: I've read the rebuttal and other reviews. I'm satisfied with the answers to my question and I maintain my recommendation for acceptance.

[Author Response · NeurIPS 2020]

We thank all reviewers for their positive comments. Below we first address common concerns among the reviewers, and then respond to questions raised by individual reviewers.

**1. Response to common concerns**

- *"Numerical experiments"*: Our paper focuses on theoretical aspects of risk-sensitive RL. It is an excellent suggestion to conduct numerical experiments to support our theoretical results. We will follow up on this.

**2. Response to individual reviewers**

**Reviewer #2**

- *"Numerical support"*: Please see our responses in the previous section.

- *"Step 10 of Alg 1"*: Yes.

- *"Non-linear functional approximator"*: At this point it is unclear how non-linear functional approximation would affect our results, and we believe that it is a very interesting and important future research direction.

**Reviewer #3**

- *"Practical relevance"*: Risk-sensitive RL finds applications in practical and strategic decision-making scenarios where risk consideration is crucial. Examples of such scenarios include, but are not limited to, autonomous driving, medicine prescriptions and financial investment. Our regret analysis provides a critical insight that under the risk-sensitive setting, the number of samples required to learn optimal policies scales exponentially in risk sensitivity, which serves as a guideline for practitioners on data collection and algorithm deployment. Our algorithms provide a way to achieve the (almost) best possible convergence rate and sample complexity for the risk-sensitive RL problem, and they are both easy to implement. Since this work focuses on theory, we leave numerical studies for our algorithms to future work.

- *"Universal constant"*: The universal constants in bonus terms are artifacts of standard concentration inequalities, and setting them to a large value such as $100$ would suffice in practice.

- *"Empirical demonstrations"*: Please see our responses in the previous section.

**Reviewer #4**

- *"Challenges of non-linearity"*: The non-linearity of the Bellman equations poses several challenges. (1) Algorithmic design: it is unclear a priori how Q-functions should be updated given the non-linear Bellman equations, and how bonus terms should be designed to enforce "optimism in the face of uncertainty" in a principled way; (2) Regret analysis: previous regret analysis of value iteration and Q-learning algorithms depends crucially on the linearity of Q-functions wrt value functions and bonus terms. It is unclear a priori how the existing proof techniques could be adapted to analyze our algorithms.

- *"Lemma 1"*: The purpose of Lemma 1 is to demonstrate a surprising contrast between the range of value functions and our regret bounds: while risk-sensitive value functions are on the same scale as their risk-neutral counterparts, which is independent of $\beta$, the regret bounds under the risk-sensitive setting have exponential dependency on $|\beta|$.

- *"Key contributions"*: Another key contribution of our work is that we provide a regret lower bound that scales exponentially in $|\beta|H$, which certifies the near optimality of our upper bounds.

- *"$b_h$"* We have defined $b_h$ in Line 9 of Alg 1.

- *"Experiments"*: Please see our responses in the previous section.

We appreciate the minor issues pointed out by the reviewers, and we will fix them in our final paper.

[Meta-Review · NeurIPS 2020]

Reviewers appreciated the important contribution towards the regret analysis of risk-sensitive reinforcement learning.